# MINORITYPROMPT: TEXT TO MINORITY IMAGE GENERATION VIA PROMPT OPTIMIZATION

## ABSTRACT

We investigate the generation of minority samples using pretrained text-to-image (T2I) latent diffusion models. Minority instances, in the context of T2I generation, can be defined as ones living on low-density regions of *text-conditional* data distributions. They are valuable for various applications of modern T2I generators, such as data augmentation and creative AI. Unfortunately, existing pretrained T2I diffusion models primarily focus on high-density regions, largely due to the influence of guided samplers (like CFG) that are essential for producing high-quality generations. To address this, we present a novel framework to counter the high-density-focus of T2I diffusion models. Specifically, we first develop an online prompt optimization framework that can encourage the emergence of desired properties during inference while preserving semantic contents of user-provided prompts. We subsequently tailor this generic prompt optimizer into a specialized solver that promotes the generation of minority features by incorporating a carefully-crafted likelihood objective. Our comprehensive experiments, conducted across various types of T2I models, demonstrate that our approach significantly enhances the capability to produce high-quality minority instances compared to existing samplers.

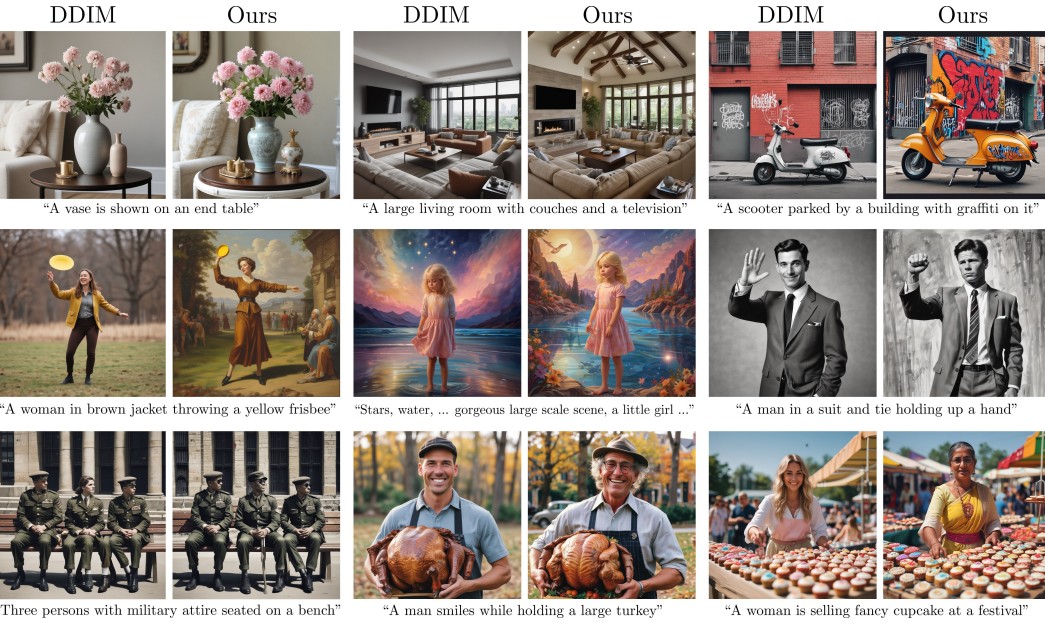

Figure 1: **Example results from our minority generation approach using SDXL-Lightning.** Our framework is designed to produce unique *minority* samples w.r.t. user-provided prompts, which are rarely generated by standard samplers like DDIM (Song et al., 2020a). Due to its low-likelihood encouraging nature, our sampler often demonstrates counteracting results against demographic biases in text-to-image models (Friedrich et al., 2023). See the samples in the last row for instance, where our sampler mitigates prevalent age and racial biases (*e.g.*, associating "man" with "young" and "woman" with "white") by modifying the demographic traits of the subjects.

# 1 INTRODUCTION

Text-to-image (T2I) generative models (Xu et al., 2018; Ramesh et al., 2021; Nichol et al., 2021) have recently attracted substantial interest for their capability to convert textual descriptions into visually striking images. At the forefront of the surge are diffusion models (Song & Ermon, 2019; Ho et al., 2020), augmented by guidance techniques (Dhariwal & Nichol, 2021; Ho & Salimans, 2022) such as classifier-free guidance (CFG) (Ho & Salimans, 2022). The guided T2I samplers encourage generations from high-density regions of a data manifold (Dhariwal & Nichol, 2021), producing realistic images that faithfully respect the provided prompts.

A key challenge is that the inherent high density focus of modern T2I samplers makes it difficult to generate *minority* samples – instances that reside in low-density regions of the manifold. This limitation is particularly significant as T2I-generated data is increasingly incorporated in downstream applications (Tian et al., 2024a;b; Afkanpour et al., 2024) where the majority-focused bias within the data may be perpetuated. Furthermore, the unique attributes found in minority instances are crucial for applications like creative AI (Rombach et al., 2022; Han et al., 2022), where generating novel and highly creative outputs is essential.

In this work, we present a novel approach dubbed as *MinorityPrompt* that counteracts the high-density bias of T2I samplers to improve their capability of minority generation. Our framework is built upon the concept of *prompt optimization*, an intuitive technique that exhibits strong performance in enhancing T2I diffusion models for various tasks (Gal et al., 2022; Chung et al., 2023b; Park et al., 2024). Unlike existing T2I-based online prompt-tuning methods that modify the entire input prompts (*e.g.*, by optimizing their text-embeddings during inference), our approach updates the prompts in a *selective* fashion to preserve the intended semantics while encouraging generations of unique low-density features. Specifically during inference, we incorporate learnable tokens into the input prompts, *e.g.*, by appending them to the end of the text. We then adjust the embeddings of these tokens across sampling timesteps, targeting the minimization of a likelihood metric designed to capture the uniqueness of noisy intermediate samples. See Figure 2 for an overview. An additional benefit of our token-based approach is that it offers enhanced semantic controllability, enabling users to express specific desired semantics in generated samples by selecting appropriate initialization words for the learnable token embeddings. To further improve the performance of our sampler, we provide new design choices that can be synergistically employed with our approach for T2I minority generation. Comprehensive experiments validate that our method can significantly improve the ability of creating minority instances of modern widely-adopted T2I models (including Stable Diffusion (SD) (Rombach et al., 2022)) with minimal compromise in sample quality and text-image alignment. In addition, we emphasize that our framework can work on distilled backbones like SDXL-Lightning (Lin et al., 2024), which demonstrates its robustness and practical relevance. As an additional application, we explore the potential of our prompt optimization framework to improve the diversity of T2I models, further exhibiting its versatility as a general-purpose solver applicable across various tasks.

Given that our prompt optimization is performed in an online manner, does not require expensive fine-tuning of T2I models, and is entirely *self-contained*, *i.e.*, implementable solely with a pretrained T2I model, we believe our approach open a new avenue for creative AI, emphasizing the practical relevance of our framework.

# 2 RELATED WORK

The generation of minority samples has been explored in a range of different scenarios and generative frameworks (Yu et al., 2020; Lin et al., 2022; Sehwag et al., 2022; Qin et al., 2023; Huang & Jafari, 2023; Um & Ye, 2023; 2024). However, significant progress has been recently made with the introduction of diffusion models, due to their ability to faithfully capture data distributions (Sehwag et al., 2022; Um & Ye, 2023; 2024). As an initial effort, Sehwag et al. (2022) incorporate separately-trained classifiers into the sampling process of diffusion models to yield guidance for low-density regions. The approach by Um & Ye (2023) shares similar intuition of integrating an additional classifier into the reverse process for low-density guidance. A limitation is that their methods rely upon external classifiers that are often difficult to obtain, especially for large-scale datasets such as T2I benchmarks (Schuhmann et al., 2022). The challenge was recently addressed by Um & Ye (2024)

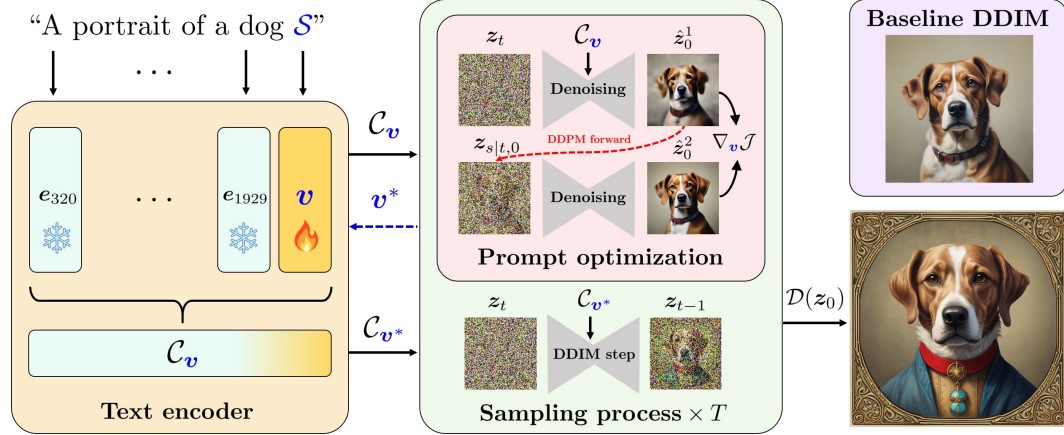

Figure 2: **Overview of MinorityPrompt.** Unlike existing online prompt tuning approaches that adjust the entire text-embedding (*e.g.*, the output of the text-encoder) during inference, our framework focuses on optimizing a dedicated *token-embedding* to better preserve the semantics within the prompt. Specifically given a user-prompt (*e.g.*, "A portrait of a dog"), we integrate a placeholder string (*e.g.*, $\mathcal{S}$ in the figure) into the prompt, marking the position of the learnable token embedding $v$. With the text-embedding $\mathcal{C}_v$ that incorporates the contents of $v$, we update $v$ *on-the-fly* during the inference process to maximize the reconstruction loss of the denoised version of $z_t$ (i.e., $\hat{z}_0^1$ in the figure). The optimized token $v^*$ is subsequently used to progress the inference at the corresponding timestep; see Section 3 for details.

where the authors develop a self-contained minority sampler that works without expensive extra components (such as classifiers). However, their method is tailored for canonical image benchmarks (like LSUN (Yu et al., 2015) and ImageNet (Deng et al., 2009)) and exhibits limited performance gain in more challenging scenarios like T2I generation. Moreover, none of these approaches have explored the dimension of prompt optimization specifically for minority generation, which is the central focus of our framework.

A related yet distinct objective is enhancing the diversity of diffusion models, an area that has been relatively overlooked compared to improving their quality. Significant progress was recently made in Sadat et al. (2023), where the authors demonstrated that adding noise perturbations, if gradually annealed over time, to conditional embeddings could greatly enhance the diversity of generated samples. However, unlike our approach, their method focuses on producing diverse samples that remain consistent with the ground-truth data distribution, rather than targeting the low-density regions of the distribution. Another notable contribution was done by Corso et al. (2023). Their idea is to repel intermediate latent samples that share the same condition, thereby encouraging the final generated samples to exhibit distinct features. A disadvantage is that it requires generating multiple instances for each prompt, which can be redundant in many practical scenarios.

Prompt optimization has been widely explored in the context of T2I diffusion models due to their strong dependence on language models. This approach has exhibited significant performance across various tasks, including inverse problems (Chung et al., 2023b) and image editing (Park et al., 2024; Mokady et al., 2023). A key difference is that most existing methods in these lines tune the *entire* prompts to find the ones that best perform the focused tasks (*e.g.*, minimizing data consistency loss (Chung et al., 2023b)). In contrast, our framework updates only the attached learnable tokens, thereby preserving the original prompt's semantics while encouraging the emergence of low-density features. Additional use cases of prompt tuning include personalization (Gal et al., 2022; Han et al., 2023) and object counting (Zafar et al., 2024). Similar to ours, their frameworks introduce variable tokens and tune their embeddings. However, their optimizations aim to learn visual concepts captured in user-provided images, whereas our focus is to invoke low-density features through optimized prompts. Also, their methods are not online, requiring separate training procedure which can be potentially expensive.

## 3 METHOD

Our focus is to generate high-quality minority instances using text-to-image (T2I) diffusion models, which faithfully reflect user-provided prompts while featuring unique visual attributes rarely produced via standard generation techniques[1]. To this end, we start with providing a brief overview on T2I diffusion frameworks and the essential background necessary to understand the core of our work. We subsequently present our proposed framework for minority generation based on the idea of prompt optimization.

### 3.1 BACKGROUND AND PRELIMINARIES

The task of T2I diffusion models is to generate an output image $x_0 \in \mathbb{R}^d$ from a random noise vector $z_T \in \mathbb{R}^k$ (where typically $k < d$), given a user-defined text prompt $\mathcal{P}$. Similar to standard (non-T2I) diffusion frameworks, the core of T2I diffusion sampling lies in an iterative denoising process that progressively removes noise from $z_T$ until a clean version $z_0$ is obtained. This denoising capability is learned through noise-prediction training (Ho et al., 2020; Song & Ermon, 2019), mathematically written as:

$$\max_{\boldsymbol{\theta}} \mathbb{E}_{z_0, y, \boldsymbol{\epsilon} \sim \mathcal{N}(\mathbf{0}, \boldsymbol{I}), t \sim \text{Unif}\{1, \ldots, T\}}[\|\boldsymbol{\epsilon} - \boldsymbol{\epsilon_\theta}(z_t, \mathcal{C})\|_2^2],$$

where $z_0 := \mathcal{E}(x_0)$, yielded by passing a training image $x_0$ through a compressive model $\mathcal{E}$ (*e.g.*, the encoder of VQ-VAE (Esser et al., 2021; Rombach et al., 2022)). Here, $z_t$ represents a noise-perturbed version of $z_0$, given by $z_t := \sqrt{\alpha_t} z_0 + \sqrt{1 - \alpha_t} \boldsymbol{\epsilon}$, where $\{\alpha_t\}_{t=1}^T$ defines the noise-schedule. $\boldsymbol{\epsilon_\theta}$ refers to a T2I diffusion model parameterized to predict the noise $\boldsymbol{\epsilon}$, and $\mathcal{C}$ represents the embedding of the text prompt $\mathcal{P}$. See below for details on how to obtain $\mathcal{C}$ from $\mathcal{P}$.

Once trained, T2I generation can be done by starting from $z_T \sim \mathcal{N}(\mathbf{0}, \boldsymbol{I})$ and implementing an iterative noise removal process guided by the text embedding $\mathcal{C}$. A common approach is to follow the deterministic DDIM sampling (Song et al., 2020a; Chung et al., 2023a):

$$z_{t-1} = \sqrt{\alpha_{t-1}} \hat{z}_0(z_t, \mathcal{C}) + \sqrt{1 - \alpha_{t-1}} \boldsymbol{\epsilon_\theta}(z_t, \mathcal{C})$$
$$\text{where} \quad \hat{z}_0(z_t, \mathcal{C}) := \frac{1}{\sqrt{\alpha_t}} \left( z_t - \sqrt{1 - \alpha_t} \boldsymbol{\epsilon_\theta}(z_t, \mathcal{C}) \right). \tag{1}$$

Here $\hat{z}_0(z_t, \mathcal{C})$ indicates a denoised estimate of $z_t$ conditioned on the text embedding $\mathcal{C}$, implemented via Tweedie's formula (Chung et al., 2022).

To further strengthen the impact of text conditioning, classifier-free guidance (CFG) (Ho & Salimans, 2022) is commonly integrated into the sampling process. In particular, one can obtain a high-density-focused noise estimation through extrapolation using an unconditional prediction:

$$\tilde{\boldsymbol{\epsilon}}_{\boldsymbol{\theta}}^w(z_t, \mathcal{C}) := w \boldsymbol{\epsilon_\theta}(z_t, \mathcal{C}) + (1 - w) \boldsymbol{\epsilon_\theta}(z_t),$$

where $\boldsymbol{\epsilon_\theta}(z_t)$ indicates an unconditional noise prediction, often implemented via null-text conditioning (Ho & Salimans, 2022). CFG refers to the technique that employs $\tilde{\boldsymbol{\epsilon}}_{\boldsymbol{\theta}}^w(z_t, \mathcal{C})$ in place of $\boldsymbol{\epsilon_\theta}(z_t, \mathcal{C})$ (in Eq. (1)), which has been shown in various scenarios to significantly improve both sample quality and text alignment yet at the expense of diversity (Sadat et al., 2023).

**Text processing.** A key distinction from non-T2I diffusion models is the incorporation of the text embedding $\mathcal{C}$, a continuous vector yielded by a text encoder $\mathcal{T}$ (such as BERT (Devlin, 2018)) based on the user prompt $\mathcal{P}$. To obtain this embedding, each word (or sub-word) in $\mathcal{P}$ is first converted into a token – an index in a pre-defined vocabulary. Each token is then mapped to a unique embedding vector through an index-based lookup. These token-wise embedding vectors, often referred to as *token* embeddings, are typically learned as part of the text encoder. The token embeddings are then passed through a transformer model, yielding the final text embedding $\mathcal{C}$. For simplicity, we denote this text processing operation as the forward pass of the text encoder $\mathcal{T}$; thus, $\mathcal{C} = \mathcal{T}(\mathcal{P})$.

**Prompt optimization.** In the context of T2I diffusion models, prompt tuning is performed by intervening in the text-processing stage. A common approach is to adjust the text embedding $\mathcal{C}$

---

[1]More formally, this can be expressed as drawing instances from $\mathcal{S}_c := \{z \in \mathcal{M}_c : p_{\boldsymbol{\theta}}(z | \mathcal{C}) < \epsilon\}$, where $\mathcal{C}$ is the prompt, $\mathcal{M}_c$ represents the (latent) data manifold associated with $\mathcal{C}$, and $p_{\boldsymbol{\theta}}$ denotes the probability density captured by the T2I diffusion model. Here $\epsilon$ is a small positive constant.

over inference time, which is widely adopted in existing online prompt optimizers (Chung et al., 2023b; Park et al., 2024). Specifically at sampling timestep $t$, existing online prompt tuners can be formulated as the following optimization problem:

$$\mathcal{C}_t^* \coloneqq \arg\max_{\mathcal{C}} \mathcal{J}(\boldsymbol{z}_t, \mathcal{C}), \tag{2}$$

where $\boldsymbol{z}_t$ is a noisy latent at step $t$ and $\mathcal{J}$ represents a task-specific objective function, such as data consistency in inverse problems (Chung et al., 2023b). Once $\mathcal{C}_t^*$ is obtained, it is used as a drop-in replacement for $\mathcal{C}$ at time $t$ (*e.g.*, in Eq. (1)), encouraging the desired property to manifest in subsequent timesteps.

A problem is that the optimization in Eq. (2) may lead to a loss of user-intended semantics in $\mathcal{P}$, due to the comprehensive updating of the entire text-embedding $\mathcal{C}$. This is critical, especially in the context of our focused T2I minority generation where preserving prompt semantics is essential; see Table 2a for our empirical results that support this. One can resort to tuning the null-text embedding while keeping $\mathcal{C}$ intact (as suggested by Mokady et al. (2023)). However, this method requires reserving the null-text dimension for this specific purpose, limiting its potential use for improving sample quality or serving other functions. In the following sections, we present an online prompt optimization framework designed to better preserve semantics. Building on this foundation, we develop our T2I minority sampler, which promotes the generation of minority features while maintaining both sample quality and text-alignment performance.

### 3.2 SEMANTIC-PRESERVING PROMPT OPTIMIZATION

The key idea of our optimization approach is to incorporate learnable tokens into a given prompt $\mathcal{P}$ and update its embedding *on-the-fly* during inference. Specifically, we append a placeholder string[2] $\mathcal{S}$ to the prompt $\mathcal{P}$, which acts as a mark for the learnable tokens. For instance, the augmented prompt could be $\mathcal{P}_{\mathcal{S}} \coloneqq$ "A portrait of a dog $\mathcal{S}$". This additional string is treated as a new vocabulary item for the text-encoder $\mathcal{T}$. We assign a token embedding $\boldsymbol{v}$ to $\mathcal{S}$, and denote the text encoder incorporating it as $\mathcal{T}(\,\cdot\,; \boldsymbol{v})$.

We propose optimizing this embedding $\boldsymbol{v}$ rather than $\mathcal{C}$. The proposed online prompt optimization at sampling step $t$ can then be formalized as follows:

$$\boldsymbol{v}_t^* \coloneqq \arg\max_{\boldsymbol{v}} \mathcal{J}(\boldsymbol{z}_t, \mathcal{C}_{\boldsymbol{v}}), \tag{3}$$

where $\mathcal{C}_{\boldsymbol{v}} \coloneqq \mathcal{T}(\mathcal{P}_{\mathcal{S}}; \boldsymbol{v})$. Afterward, the optimized text-embedding $\mathcal{C}_{\boldsymbol{v}_t^*}$ is obtained by text-processing $\mathcal{P}_{\mathcal{S}}$ with the updated token-embedding of $\mathcal{S}$, therefore $\mathcal{C}_{\boldsymbol{v}_t^*} \coloneqq \mathcal{T}(\mathcal{P}_{\mathcal{S}}; \boldsymbol{v}_t^*)$.

Note that our optimization does not affect the embeddings of the tokens w.r.t. the original prompt $\mathcal{P}$. This is inherently more advantageous for preserving semantics compared to existing methods, which alter the entire text-embedding $\mathcal{C}$ and thereby effectively impact all token embeddings. We also highlight that unlike existing learnable-token-based approaches that share the same embedding throughout inference (Gal et al., 2022; Han et al., 2023; Zafar et al., 2024), our framework allows the token embedding $\boldsymbol{v}$ to change over timesteps $t$. This adaptive feature offers potential advantages, since the role of $\boldsymbol{v}$ in maximizing $\mathcal{J}$ can vary with the changing nature of $\boldsymbol{z}_t$ across different timesteps. This point is also implied in previous works that employ adaptive text-embeddings over time (Chung et al., 2023b; Park et al., 2024).

Intuitively, our optimization can be understood as capturing a specific concept relevant to noisy latent $\boldsymbol{z}_t$ within the token $\boldsymbol{v}_t^*$, guided by the objective function $\mathcal{J}$. Thanks to its general design that accommodates any arbitrary objective function $\mathcal{J}$, this framework is versatile and can be employed in various contexts beyond minority generation. For instance, it can be used to diversify the outputs of T2I models. See details in Table 3b.

### 3.3 MINORITYPROMPT: MINORITY-FOCUSED PROMPT TUNING

We now specialize the generic solver in Eq. (3) for the task of minority generation. The key question is how to formulate an appropriate objective function $\mathcal{J}$ for this purpose. To address this, we

---

[2]The placeholder string can be placed at any position in the prompt, but we empirically found that inserting it at the end of the prompt yields the best performance; see Table 6b for details.

| **Algorithm 1** MinorityPrompt | **Algorithm 2** Prompt optimization |
|---|---|
| **Require:** $\epsilon_{\boldsymbol{\theta}}, \mathcal{T}, \mathcal{D}, \boldsymbol{v}_T^{(0)}, \mathcal{P}_{\mathcal{S}}, \mathcal{C}, N, K, w, T, s, \lambda$. | 1: **function** OPTIMIZEEMB($\boldsymbol{z}_t, \boldsymbol{v}_t^{(0)}, \epsilon_{\boldsymbol{\theta}}, \mathcal{T}, K, s, \lambda$) |
| 1: $\boldsymbol{z}_T \sim \mathcal{N}(\boldsymbol{0}, \boldsymbol{I})$ | 2:     **for** $k \leftarrow 1$ to $K$ **do** |
| 2: **for** $t \leftarrow T$ to 1 **do** | 3:        $\mathcal{C}_{\boldsymbol{v}} \leftarrow \mathcal{T}(\mathcal{P}_{\mathcal{S}}; \boldsymbol{v}_t^{(k-1)})$ |
| 3:    $\mathcal{C}_{\boldsymbol{v}_t^*} \leftarrow \mathcal{C}$ | 4:        $\epsilon_{\boldsymbol{\theta}}^1 \leftarrow \epsilon_{\boldsymbol{\theta}}(\boldsymbol{z}_t, \mathcal{C}_{\boldsymbol{v}})$ |
| 4:    **if** $t \bmod N = 0$ **then** | 5:        $\hat{\boldsymbol{z}}_0^1 \leftarrow (\boldsymbol{z}_t - \sqrt{1-\alpha_t}\epsilon_{\boldsymbol{\theta}}^1)/\sqrt{\alpha_t}$ |
| 5:      $\boldsymbol{v}_t^* \leftarrow$ OPTIMIZEEMB($\boldsymbol{z}_t, \boldsymbol{v}_t^{(0)}, \epsilon_{\boldsymbol{\theta}}, \mathcal{T}, K, s, \lambda$) | 6:        $\epsilon \sim \mathcal{N}(\boldsymbol{0}, \boldsymbol{I})$ |
| 6:      $\mathcal{C}_{\boldsymbol{v}_t^*} \leftarrow \mathcal{T}(\mathcal{P}_{\mathcal{S}}; \boldsymbol{v}_t^*)$ | 7:        $\boldsymbol{z}_{s|t,0} \leftarrow \sqrt{\alpha_s}\hat{\boldsymbol{z}}_0^1 + \sqrt{1-\alpha_s}\epsilon$ |
| 7:    **end if** | 8:        $\epsilon_{\boldsymbol{\theta}}^2 \leftarrow \epsilon_{\boldsymbol{\theta}}(\boldsymbol{z}_{s|t,0}, \mathcal{C})$ |
| 8:    $\tilde{\epsilon}_{\boldsymbol{\theta}}^w \leftarrow w\epsilon_{\boldsymbol{\theta}}(\boldsymbol{z}_t, \mathcal{C}_{\boldsymbol{v}_t^*}) + (1-w)\epsilon_{\boldsymbol{\theta}}(\boldsymbol{z}_t)$ | 9:        $\hat{\boldsymbol{z}}_0^2 \leftarrow (\boldsymbol{z}_{s|t,0} - \sqrt{1-\alpha_s}\epsilon_{\boldsymbol{\theta}}^2)/\sqrt{\alpha_s}$ |
| 9:    $\hat{\boldsymbol{z}}_0^w \leftarrow (\boldsymbol{z}_t - \sqrt{1-\alpha_t}\tilde{\epsilon}_{\boldsymbol{\theta}}^w)/\sqrt{\alpha_t}$ | 10:      $\mathcal{J}_t \leftarrow \|\hat{\boldsymbol{z}}_0^1 - \text{sg}(\hat{\boldsymbol{z}}_0^2)\|_2^2 + \lambda\|\text{sg}(\hat{\boldsymbol{z}}_0^1) - \hat{\boldsymbol{z}}_0^2\|_2^2$ |
| 10:   $\boldsymbol{z}_{t-1} \leftarrow \sqrt{\alpha_{t-1}}\hat{\boldsymbol{z}}_0^w + \sqrt{1-\alpha_{t-1}}\tilde{\epsilon}_{\boldsymbol{\theta}}^w$ | 11:      $\boldsymbol{v}_t^{(k)} \leftarrow \boldsymbol{v}_t^{(k-1)} + \text{AdamGrad}(\mathcal{J}_t)$ |
| 11:   $\boldsymbol{v}_{t-1}^{(0)} \leftarrow \boldsymbol{v}_t^*$ | 12:     **end for** |
| 12: **end for** | 13:     **return** $\boldsymbol{v}_t^* \leftarrow \boldsymbol{v}_t^{(K)}$ |
| 13: **return** $\boldsymbol{x}_0 \leftarrow \mathcal{D}(\boldsymbol{z}_0)$ | 14: **end function** |

draw inspiration from Um & Ye (2024), employing their likelihood metric as the starting point for developing our objective function.

Since the metric was originally defined in the pixel domain using non-T2I diffusion models (see Section B.1 for details), we initially perform a naive adaptation to accommodate the latent space of interest, $\boldsymbol{z}_t \in \mathbb{R}^k$, and integrate text conditioning using CFG as is typical in the T2I context (Kim et al., 2023). The adapted version of the metric reads:

$$\mathcal{J}(\boldsymbol{z}_t, \mathcal{C}) := \mathbb{E}_{\epsilon}\big[\|\hat{\boldsymbol{z}}_0^w(\boldsymbol{z}_t, \mathcal{C}) - \text{sg}(\hat{\boldsymbol{z}}_0^w(\boldsymbol{z}_{s|t,0}^w, \mathcal{C}))\|_2^2\big], \tag{4}$$

where $\hat{\boldsymbol{z}}_0^w(\boldsymbol{z}_t, \mathcal{C})$ represents a clean estimate of $\boldsymbol{z}_t$ using the CFG noise term $\tilde{\epsilon}_{\boldsymbol{\theta}}^w(\boldsymbol{z}_t, \mathcal{C})$. Here $\boldsymbol{z}_{s|t,0}^w$ indicates a noised version of $\hat{\boldsymbol{z}}_0^w(\boldsymbol{z}_t, \mathcal{C})$ w.r.t. timestep $s$: $\boldsymbol{z}_{s|t,0}^w := \sqrt{\alpha_s}\hat{\boldsymbol{z}}_0^w(\boldsymbol{z}_t, \mathcal{C}) + \sqrt{1-\alpha_s}\epsilon$, and $\hat{\boldsymbol{z}}_0^w(\boldsymbol{z}_{s|t,0}^w, \mathcal{C})$ is a clean version of $\boldsymbol{z}_{s|t,0}^w$ conditioned on $\mathcal{C}$. $\text{sg}(\cdot)$ denotes the stop-gradient operator for reducing computational cost when used in guided sampling (Um & Ye, 2024). Notice that the squared L2 error is used as the discrepancy loss, rather than the originally used LPIPS (Zhang et al., 2018), due to its incompatibility with our latent space. The quantity in Eq. (4) is interpretable as a reconstruction loss of $\hat{\boldsymbol{z}}_0^w(\boldsymbol{z}_t, \mathcal{C})$. As exhibited in Um & Ye (2024), the loss may become large if $\boldsymbol{z}_t$ (represented by $\hat{\boldsymbol{z}}_0^w(\boldsymbol{z}_t, \mathcal{C})$) contains highly-unique minority features that often vanish during the reconstruction process. The comprehensive details regarding the original metric due to Um & Ye (2024) are provided in Section B.1.

Considering Eq. (4) as the objective function, a natural approach for minority-focused prompt tuning would be to incorporate $\mathcal{C}_{\boldsymbol{v}}$ and optimize for the best $\boldsymbol{v}$:

$$\boldsymbol{v}_t^* := \arg\max_{\boldsymbol{v}} \mathcal{J}(\boldsymbol{z}_t, \mathcal{C}_{\boldsymbol{v}})$$
$$\text{where} \quad \mathcal{J}(\boldsymbol{z}_t, \mathcal{C}_{\boldsymbol{v}}) := \mathbb{E}_{\epsilon}\big[\|\hat{\boldsymbol{z}}_0^w(\boldsymbol{z}_t, \mathcal{C}_{\boldsymbol{v}}) - \text{sg}(\hat{\boldsymbol{z}}_0^w(\boldsymbol{z}_{s|t,0}^w, \mathcal{C}_{\boldsymbol{v}}))\|_2^2\big]. \tag{5}$$

However, we argue that this naively extended framework has theoretical issues that lead to limited performance gain over standard samplers. Specifically, three aspects of this objective weaken the desired connection to the target log-likelihood $\log p_{\boldsymbol{\theta}}(\boldsymbol{z}_0 \mid \mathcal{C})$ that we aim to capture: (i) the reliance on the CFG-based clean predictions; (ii) obstructed gradient flow through the second term in the squared-L2 loss; and (iii) the incorporation of $\mathcal{C}_{\boldsymbol{v}}$ within the second term in the loss. See Section A.2 on a detailed analysis on these points.

Hence, we propose the following optimization to address the theoretical issues:

$$\boldsymbol{v}_t^* := \arg\max_{\boldsymbol{v}} \mathcal{J}_{\mathcal{C}}(\boldsymbol{z}_t, \mathcal{C}_{\boldsymbol{v}})$$
$$\text{where} \quad \mathcal{J}_{\mathcal{C}}(\boldsymbol{z}_t, \mathcal{C}_{\boldsymbol{v}}) := \mathbb{E}_{\epsilon}\big[\|\hat{\boldsymbol{z}}_0(\boldsymbol{z}_t, \mathcal{C}_{\boldsymbol{v}}) - \hat{\boldsymbol{z}}_0(\boldsymbol{z}_{s|t,0}, \mathcal{C})\|_2^2\big] \tag{6}$$

where $\hat{\boldsymbol{z}}_0(\boldsymbol{z}_t, \mathcal{C}_{\boldsymbol{v}}) := (\boldsymbol{z}_t - \sqrt{1-\alpha_t}\epsilon_{\boldsymbol{\theta}}(\boldsymbol{z}_t, \mathcal{C}_{\boldsymbol{v}}))/\sqrt{\alpha_t}$, indicating a non-CFG clean estimate. We found that the proposed optimization maintains a close connection to the focused log-likelihood. Below we provide a formal statement of our finding. See Section A.1 for the proof.

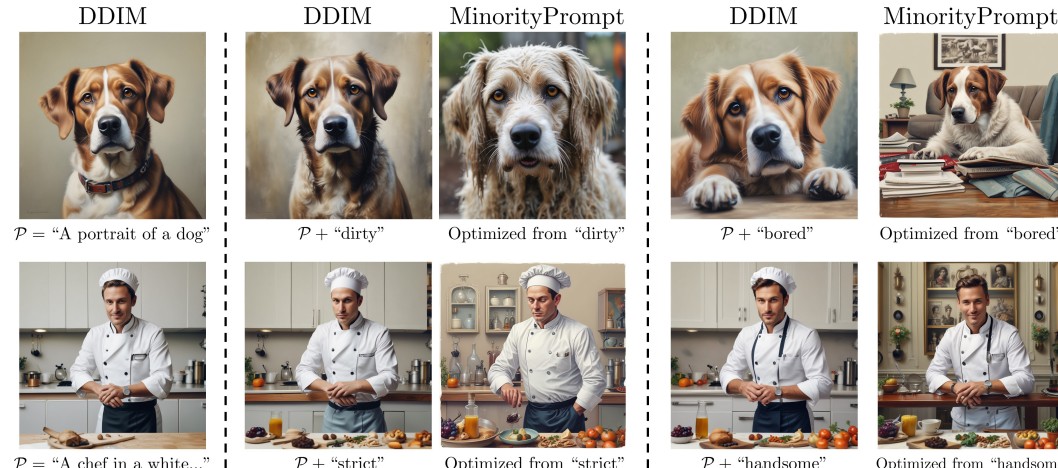

Figure 3: **Enhanced semantic controllability by MinorityPrompt.** The samples in the first column are generations due to DDIM using the two base prompts (*e.g.*, "A portrait of a dog" for the first row). The third and fifth columns exhibit generated samples from our framework, where we selected the corresponding word embeddings as the starting points of the prompt optimizations. For comparison, we also present DDIM samples produced using attached prompts with the corresponding words in the the second and fourth columns. For instance in the first row, the image in the second column corresponds to the generation due to "A portrait of a dog *dirty*". All samples were obtained using SDXL-Lightning (Lin et al., 2024)

**Proposition 1.** *The objective function in Eq. (6) is equivalent (upto a constant factor) to the negative ELBO w.r.t.* $\log p_{\boldsymbol{\theta}}(\hat{\boldsymbol{z}}_0(\boldsymbol{z}_t, \mathcal{C}_{\boldsymbol{v}}) \mid \mathcal{C})$ *when integrated over timesteps with* $\bar{w}_s := \alpha_s/(1 - \alpha_s)$:

$$\sum_{s=1}^{T} \bar{w}_s \mathcal{J}_{\mathcal{C}}(\boldsymbol{z}_t, \mathcal{C}_{\boldsymbol{v}}) = \sum_{s=1}^{T} \mathbb{E}_{\boldsymbol{\epsilon}}[\|\boldsymbol{\epsilon} - \boldsymbol{\epsilon}_{\boldsymbol{\theta}}(\sqrt{\alpha_s}\hat{\boldsymbol{z}}_0(\boldsymbol{z}_t, \mathcal{C}_{\boldsymbol{v}}) + \sqrt{1-\alpha_s}\boldsymbol{\epsilon}, \mathcal{C})\|_2^2]$$

$$\gtrless -\log p_{\boldsymbol{\theta}}(\hat{\boldsymbol{z}}_0(\boldsymbol{z}_t, \mathcal{C}_{\boldsymbol{v}}) \mid \mathcal{C}).$$

Intuitively, our optimization seeks to make the text-conditioned clean view $\hat{\boldsymbol{z}}_0(\boldsymbol{z}_t, \mathcal{C}_{\boldsymbol{v}})$ of the current sample $\boldsymbol{z}_t$ as unique as possible, from the perspective of the log-likelihood $\log p_{\boldsymbol{\theta}}(\hat{\boldsymbol{z}}_0(\boldsymbol{z}_t, \mathcal{C}_{\boldsymbol{v}})|\mathcal{C})$.

**Techniques for improvement.** In practice, we found that our optimization could be further stabilized by introducing a `sg`-related trick into the objective function:

$$\tilde{\mathcal{J}}_{\mathcal{C}} := \mathcal{J}_{\mathcal{C}}^1 + \lambda \mathcal{J}_{\mathcal{C}}^2, \quad \lambda > 0$$

$$\text{where} \quad \mathcal{J}_{\mathcal{C}}^1 := \mathbb{E}_{\boldsymbol{\epsilon}}\left[\left\|\hat{\boldsymbol{z}}_0(\boldsymbol{z}_t, \mathcal{C}_{\boldsymbol{v}}) - \texttt{sg}\left(\hat{\boldsymbol{z}}_0(\boldsymbol{z}_{s|t,0}, \mathcal{C})\right)\right\|_2^2\right] \quad (7)$$

$$\mathcal{J}_{\mathcal{C}}^2 := \mathbb{E}_{\boldsymbol{\epsilon}}\left[\left\|\texttt{sg}\left(\hat{\boldsymbol{z}}_0(\boldsymbol{z}_t, \mathcal{C}_{\boldsymbol{v}})\right) - \hat{\boldsymbol{z}}_0(\boldsymbol{z}_{s|t,0}, \mathcal{C})\right\|_2^2\right].$$

In our empirical results, setting $\lambda = 1$ consistently produces the best performance across all considered T2I models. We note that this technique allows the gradient flow through the second term (contrary to the case of Eq. (5)), thereby sidestepping the gradient blocking issue that we mentioned earlier. Another significant improvement comes from the use of an annealed timestep $s$, which was originally adhered to a fixed value in Um & Ye (2024). We empirically found that employing an annealing schedule based on the inverse of the sampling step (*e.g.*, $s = T - t$) outperforms other fixed choices of $s$. Similar to Um & Ye (2024), we conduct our prompt optimization intermittently (*i.e.*, once every $N$ sampling steps) to reduce computational costs. We found that during non-optimizing steps, employing the base prompt $\mathcal{C}$ instead of $\mathcal{C}_{\boldsymbol{v}}$ (with the most recently updated token embedding) yields improvements in text-alignment and sample quality. See Algorithms 1 and 2 for the pseudocode of our approach.

**Enhanced semantic controllability.** A key benefit of our prompt optimization approach is its ability to provide an additional dimension of semantic control over the generated samples. Specifically, by selecting an appropriate initial point for $\boldsymbol{v}$ (*i.e.*, $\boldsymbol{v}_T^{(0)}$ in Algorithm 1), such as a word embedding

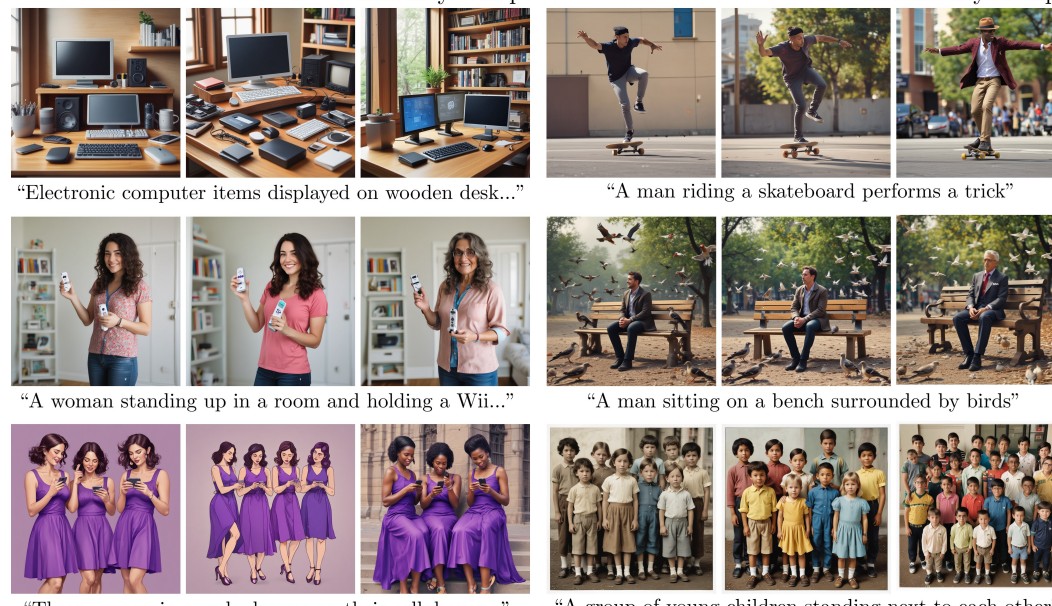

DDIM     SGMS     MinorityPrompt       DDIM     SGMS     MinorityPrompt

"Electronic computer items displayed on wooden desk..."     "A man riding a skateboard performs a trick"

"A woman standing up in a room and holding a Wii..."     "A man sitting on a bench surrounded by birds"

"Three women in purple dresses on their cellphones..."     "A group of young children standing next to each other"

Figure 4: **Sample comparison on SDXL-Lightning.** Generated samples from three different approaches: (i) DDIM (Song et al., 2020a); (ii) SGMS (Um & Ye, 2024); (iii) MinorityPrompt (ours). Six distinct prompts were used for this comparison, and random seeds were shared across all three methods.

with relevant semantics, one can impart the desired semantics to the generated output; see Figure 3 for instance. Note that the controllability is not achievable with existing minority samplers that rely upon latent-space optimizations (Sehwag et al., 2022; Um & Ye, 2023; 2024). In addition, we found that properly choosing a initial word can yield improved minority generation performance compared to approaches that rely upon random starting points; see Table 2c for details.

## 4 EXPERIMENTS

### 4.1 SETUP

**T2I backbones and dataset.** Our experiments were conducted using three distinct versions of Stable Diffusion (SD) (Rombach et al., 2022), encompassing both standard and distilled versions to demonstrate the robustness of our approach. Specifically, we consider: (i) SDv1.5; (ii) SDv2.0; (iii) SDXL-Lightning (SDXL-LT) (Lin et al., 2024). For all pretrained models, we employed the widely-adopted HuggingFace checkpoints trained on LAION (Schuhmann et al., 2022) without any further modifications. As convention, we randomly selected 10K captions from the validation set of MS-COCO (Lin et al., 2014) for the generations with SDv1.5 and SDv2.0 while using 5K captions for the SDXL-Lightning results.

**Baselines.** The same four baselines were considered over all SD versions: (i) the standard DDIM (Song et al., 2020a); (ii) a null-prompted DDIM; (iii) CADS (Sadat et al., 2023); (iv) SGMS (Um & Ye, 2024). The null-prompted DDIM serves as a naive baseline that attempts to encourage unique sampling by incorporating a proper null-text prompt, such as "commmonly-looking". CADS (Sadat et al., 2023) is the state-of-the-art diversity-focused sampler that may rival our approach in minority generation, while SGMS (Um & Ye, 2024) is the state-of-the-art of minority generation outside the T2I domain. We adhered to standard sampling setups for all methods. Specifically, 50 DDIM steps (*i.e.*, $T = 50$) with $w = 7.5$ were used for SDv1.5 and SDv2.0, while $w = 1.0$ was employed for the 4-step SDXL-Lightning model.

**Evaluations.** For evaluating text-alignment and user-preference, we consider three distinct quantities: (i) ClipScore (Hessel et al., 2021); (ii) PickScore (Kirstain et al., 2023); (iii) Image-Reward (Xu et al., 2023). We additionally employ two metrics for quality and diversity: Precision and Re-

| Model | Method | CLIPScore ↑ | PickScore ↑ | ImageReward ↑ | Precision ↑ | Recall ↑ | Likelihood ↓ |
|---|---|---|---|---|---|---|---|
| SDv1.5 | DDIM | 31.4801 | 21.4830 | 0.2106 | **0.5907** | 0.6328 | 1.0367 |
| | DDIM + null | 31.1007 | **21.5391** | **0.2422** | 0.5660 | 0.6236 | 1.0339 |
| | CADS | 31.4178 | 21.2836 | 0.1012 | 0.5696 | **0.6346** | 1.0127 |
| | SGMS | 31.1665 | 21.2126 | 0.1230 | 0.4943 | 0.5960 | 0.9540 |
| | MinorityPrompt | **31.5376** | 21.3111 | 0.2352 | 0.5671 | 0.6228 | **0.8971** |
| SDv2.0 | DDIM | 31.8490 | 21.6801 | 0.3821 | 0.5930 | 0.6292 | 1.1100 |
| | DDIM + null | 31.7223 | **21.7190** | 0.4024 | 0.5861 | **0.6308** | 1.0769 |
| | CADS | 31.7687 | 21.5225 | 0.2981 | 0.5811 | 0.6194 | 1.0851 |
| | SGMS | 31.4750 | 21.4457 | 0.2981 | 0.5166 | 0.6130 | 0.9898 |
| | MinorityPrompt | **31.9586** | 21.5958 | **0.4249** | **0.6047** | 0.6100 | **0.9143** |
| SDXL-LT | DDIM | 31.5714 | 22.6822 | 0.7317 | **0.5306** | 0.6648 | 0.6102 |
| | DDIM + null | **31.5754** | **22.7124** | **0.7397** | 0.5194 | 0.6602 | 0.6093 |
| | CADS | 31.0837 | 22.3690 | 0.4946 | 0.5244 | 0.6594 | 0.6038 |
| | SGMS | 31.3589 | 22.5866 | 0.6759 | 0.4868 | **0.6968** | 0.5470 |
| | MinorityPrompt | 31.3838 | 22.6157 | 0.7042 | 0.4758 | 0.6928 | **0.5463** |

Table 1: **Quantitative comparisons.** "SDXL-LT" denotes SDXL-Lightning (4-step version) (Lin et al., 2024). "DDIM + null" indicates a baseline that leverages a properly-chosen null-prompt to encourage minority generations, where we used "commonly-looking" for the results herein. "CADS (Sadat et al., 2023)" is the state-of-the-art in diverse sampling, while SGMS (Um & Ye, 2024) denotes a minority sampler similar to ours, representing the state-of-the-art outside the T2I context. "Likelihood" represents log-likelihood values measured in bpd (bits per dimension).

call (Kynkäänniemi et al., 2019). For the likelihood of generated samples, we rely upon the exact likelihood computation method based on PF-ODE as proposed by Song et al. (2020b). Notably, we do not include Fréchet Inception Distance (FID) (Heusel et al., 2017) as an evaluator, since FID measures closeness to baseline real data (*e.g.*, the MS-COCO validation set), which diverges from our focus on promoting generations in low-density regions.

## 4.2 RESULTS

**Qualitative comparisons.** Figure 4 presents a comparison of generated samples of our approach with two baselines. Notice that our MinorityPrompt tends to yield highly more distinct and complex features (*e.g.*, intricate visual elements (Arvinte et al., 2023; Serrà et al., 2019)) compared to the baseline samplers, demonstrating its effectiveness even with distilled pretrained models. A significant observation, also reflected in Figure 1, is that MinorityPrompt often counters the inherent demographic biases of T2I models, *e.g.*, by adjusting age or skin color. See the samples in the second and third rows of the figure. A more extensive set of generated samples, including those from SDv1.5 and v2.0, can be found in Section D.2.

**Quantitative evaluations.** Table 1 exhibits performance comparisons across three distinct T2I models. Observe that our sampler outperforms all baselines in generating low-likelihood samples, while maintaining reasonable performance in text-to-alignment and user preference; also see Figure 5 where we present the distributions of log-likelihood. However, the performance trends for SDXL-LT results differ slightly from those of SDv1.5 and SDv2.0 across all tailored samplers, with particularly degraded results. We attribute this to the small-step nature of distilled models, which offer fewer opportunities to intervene in the sampling process, thereby limiting the potential for quantitative improvements.

**Ablation studies.** Table 2 investigates the impact of key design choices in our framework. Specifically, Table 2a highlights the benefits of optimizing small sets of token embeddings, which outperform alternatives targeting text or null-text embeddings in both text alignment and log-likelihood. The advantage of using the proposed objective function Eq. (6) is exhibited in Table 2b, where the naively-extended framework based on Eq. (5) demonstrates significant performance gap compared to our carefully-crafted approach. Table 2c explores various initialization techniques for $v$. While all methods yield substantial improvements over the unoptimized sampler (see "unoptimized" in Table 2b for comparison), we observe that further gains can be achieved with properly chosen initial words. A more comprehensive analysis and ablation study, encompassing additional design choices and applications to trending sampling techniques such as CFG++ (Chung et al., 2024), is presented in Section C.1.

| Target | CS ↑ | LL ↓ |
|---|---|---|
| Text | 31.3503 | 0.9263 |
| Null-text | 31.1089 | 1.0175 |
| Token (ours) | **31.6465** | **0.9006** |

| Method | CS ↑ | LL ↓ |
|---|---|---|
| Unoptimized | 31.4395 | 1.0465 |
| Naive (Eq. (5)) | 30.2994 | 0.9245 |
| Ours (Eq. (6)) | **31.7369** | **0.9230** |

| Type | CS ↑ | LL ↓ |
|---|---|---|
| Default | 31.5154 | 0.9355 |
| Gaussian | 31.5054 | 0.9429 |
| Word init | **31.7369** | **0.9230** |

(a) Influence of optimization target | (b) Impact of objective function $\mathcal{J}$ | (c) Effect of initializing $v$

Table 2: **Ablation study results.** "CS" denotes ClipScore (Hessel et al., 2021), while 'LL' indicates log-likelihood. "Text" is the optimization framework focused on updating the text-embedding $\mathcal{C}$, and "Null-text" refers to the one that adjusts the null-text embedding (as in (Mokady et al., 2023)). "Unoptimized" corresponds to the standard DDIM sampler. "Default" denotes the case that simply employs the default embedding assigned with an added learnable token, while "Gaussian" initializes $v$ from a multivariate Gaussian distribution constructed using the mean and variance of the token embeddings from the text-encoder $\mathcal{T}$. "Word init" indicates initializing with a specific word embedding. We used SDv1.5 for the results herein.

**Further applications.** Beyond our primary focus on minority generation, we additionally investigate the potential of our framework, specifically in the perspectives of fairness and diversity. Table 3a presents one such instance. Although not explicitly designed to address demographic biases, our minority sampler demonstrates the ability to counteract gender bias and produce more neutral generation results. This corroborates with our qualitative observations made in Figures 1 and 4.

Another area of investigation involves diversity, where we validate the versatility of our prompt optimizer in Eq. (3) for fostering diverse generation. To achieve this, we develop a new objective function aimed at encouraging diversity within a sampling batch that shares the same prompt (*i.e.*, similar to the goal in Corso et al. (2023)) by enforcing repulsion between generated instances:

$$\bar{\mathcal{J}} := \sum_{i=1}^{\mathcal{B}} \sum_{k \neq i} \|\hat{z}_0(z_t^{(i)}, \mathcal{C}_v) - \hat{z}_0(z_t^{(j)}, \mathcal{C}_v)\|_2^2, \tag{8}$$

where $\mathcal{B}$ is the batch size, and $\{z_t^{(i)}\}_{i=1}^{\mathcal{B}}$ denotes the noisy instances in the batch. We found that incorporating this objective into Eq. (3) yields impressive results, even rivaling the state-of-the-art diverse sampler (Sadat et al., 2023); see Table 3b for details.

| Method | $|p_{\text{female}} - p_{\text{male}}|$ ↓ |
|---|---|
| DDIM | 0.3600 |
| CADS | 0.2686 |
| Ours | **0.2342** |

| Method | ClipScore ↑ | PickScore ↑ | ImReward ↑ | Prec ↑ | Rec ↑ | IBS ↓ |
|---|---|---|---|---|---|---|
| DDIM | **31.4393** | **21.2478** | **0.0121** | **0.5860** | **0.6390** | 0.6164 |
| CADS | 31.2692 | 21.0262 | -0.0976 | 0.5620 | 0.5980 | 0.5494 |
| Ours | 31.2724 | 21.0404 | -0.1204 | 0.5480 | 0.6316 | **0.5439** |

(a) Debiasing effect of ours | (b) Effectiveness of our diversity-focused framework in Eq. (8)

Table 3: **(a) Bias-mitigating impact of MinorityPrompt.** $p_{\text{female}}$ indicates the proportion of females in generated samples via gender-neutral prompts that include "a person" (*e.g.*, "A person doing karate in a field at night"). On the other hand, $p_{\text{male}}$ is the proportion of males in the samples. We employed SDXL-LT for these results. **(b) Effectiveness of our diversity-focused prompt optimization framework**. "ImReward" denotes Image-Reward metric. "IBS" represents In-Batch Similarity, a diversity metric (Corso et al., 2023) that evaluates the cosine similarity in the DINO feature space (Caron et al., 2021). The results were obtained on SDv1.5.

## 5 CONCLUSION

We developed a novel framework for generating minority samples in the context of T2I generation. Built upon our prompt optimization framework that updates the embeddings of additional learnable tokens, our minority sampler offers significant performance improvements both in text-alignment and low-likelihood generation compared to existing approaches. To accomplish this, we meticulously tailor the objective function with theoretical justifications and implement several techniques for further enhancements. Beyond our main interest of minority generation, we further demonstrated the potential of our framework in promoting fairness and diversity. During this process, we also showed that the proposed optimization framework can serve as a general solution, with potential applicability to various optimization tasks associated with T2I generation.

ETHICS STATEMENT

One potential concern associated with our approach is the possibility of its malicious use to inhibit the generation of minority-featured samples. For instance, this could occur by flipping the sign of the objective function Eq. (6), yielding a focus on high-density generations. It is crucial to recognize this risk and to ensure that our proposed framework is employed responsibly to foster fairness and inclusivity in generative modeling.

REPRODUCIBILITY

To ensure the reproducibility of our experiments, we provide a comprehensive description regarding the employed pretrained models for our experiments. All experimental settings, including hyper-parameter choices, are detailed in Section B.2. Additionally, we include the average running time of our algorithm along with specific details about the computer configuration in the same section. Finally, to assist with replication efforts, we have made our code available in a public repository: https://github.com/anonymous-6898/MinorityPrompt.

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

## A    THEORETICAL RESULTS

### A.1    PROOF OF PROPOSITION 1

**Proposition 1.** *The objective function in Eq. (6) is equivalent (upto a constant factor) to the negative ELBO w.r.t.* $\log p_{\boldsymbol{\theta}}(\hat{z}_0(z_t, \mathcal{C}_{\boldsymbol{v}}) \mid \mathcal{C})$ *when integrated over timesteps with* $\bar{w}_s := \alpha_s/(1-\alpha_s)$:

$$\sum_{s=1}^{T} \bar{w}_s \mathcal{J}_{\mathcal{C}}(z_t, \mathcal{C}_{\boldsymbol{v}}) = \sum_{s=1}^{T} \mathbb{E}_{\boldsymbol{\epsilon}}[\|\boldsymbol{\epsilon} - \boldsymbol{\epsilon}_{\boldsymbol{\theta}}(\sqrt{\alpha_s}\hat{z}_0(z_t, \mathcal{C}_{\boldsymbol{v}}) + \sqrt{1-\alpha_s}\boldsymbol{\epsilon}, \mathcal{C})\|_2^2] \qquad (9)$$
$$\gtrapprox -\log p_{\boldsymbol{\theta}}(\hat{z}_0(z_t, \mathcal{C}_{\boldsymbol{v}}) \mid \mathcal{C}).$$

*Proof.* Remember the definition of the objective function in Eq. (6):

$$\mathcal{J}_{\mathcal{C}}(z_t, \mathcal{C}_{\boldsymbol{v}}) := \mathbb{E}_{\boldsymbol{\epsilon}}\big[\|\hat{z}_0(z_t, \mathcal{C}_{\boldsymbol{v}}) - \hat{z}_0(z_{s|t,0}, \mathcal{C})\|_2^2\big].$$

Plugging this into the LHS of Eq. (9) yields:

$$\sum_{s=1}^{T} \bar{w}_s \mathcal{J}_{\mathcal{C}}(z_t, \mathcal{C}_{\boldsymbol{v}}) = \sum_{s=1}^{T} \frac{\alpha_s}{1-\alpha_s} \mathbb{E}_{\boldsymbol{\epsilon}}\big[\|\hat{z}_0(z_t, \mathcal{C}_{\boldsymbol{v}}) - \hat{z}_0(z_{s|t,0}, \mathcal{C})\|_2^2\big]$$

$$= \sum_{s=1}^{T} \frac{\alpha_s}{1-\alpha_s} \mathbb{E}_{\boldsymbol{\epsilon}} \left[\left\|\frac{1}{\sqrt{\alpha_s}}(z_{s|t,0} - \sqrt{1-\alpha_s}\boldsymbol{\epsilon}) - \frac{1}{\sqrt{\alpha_s}}(z_{s|t,0} - \sqrt{1-\alpha_s}\boldsymbol{\epsilon}_{\boldsymbol{\theta}}(z_{s|t,0}, \mathcal{C}))\right\|_2^2\right]$$

$$= \sum_{s=1}^{T} \frac{\alpha_s}{1-\alpha_s} \mathbb{E}_{\boldsymbol{\epsilon}} \left[\left\|\frac{\sqrt{1-\alpha_s}}{\sqrt{\alpha_s}}(\boldsymbol{\epsilon} - \boldsymbol{\epsilon}_{\boldsymbol{\theta}}(z_{s|t,0}, \mathcal{C}))\right\|_2^2\right] \qquad (10)$$

$$= \sum_{s=1}^{T} \mathbb{E}_{\boldsymbol{\epsilon}}[\|\boldsymbol{\epsilon} - \boldsymbol{\epsilon}_{\boldsymbol{\theta}}(\sqrt{\alpha_s}\hat{z}_0(z_t, \mathcal{C}_{\boldsymbol{v}}) + \sqrt{1-\alpha_s}\boldsymbol{\epsilon}, \mathcal{C})\|_2^2],$$

where the second equality is from the definitions of $z_{s|t,0}$ and $\hat{z}_0(z_{s|t,0}, \mathcal{C})$:

$$z_{s|t,0} := \sqrt{\alpha_s}\hat{z}_0(z_t, \mathcal{C}_{\boldsymbol{v}}) + \sqrt{1-\alpha_s}\boldsymbol{\epsilon}$$
$$\hat{z}_0(z_{s|t,0}, \mathcal{C}) := \frac{1}{\sqrt{\alpha_s}}(z_{s|t,0} - \sqrt{1-\alpha_s}\boldsymbol{\epsilon}_{\boldsymbol{\theta}}(z_{s|t,0}, \mathcal{C})).$$

Note that the last expression in Eq. (10), which is the same as the RHS of Eq. (9), is equivalent (up to a constant) to the expression of the negative ELBO w.r.t. $\hat{z}_0(z_t, \mathcal{C}_{\boldsymbol{v}})$ (Ho et al., 2020; Li et al., 2023). The distinction here is that now we use a text-conditional diffusion model $\boldsymbol{\epsilon}_{\boldsymbol{\theta}}(\cdot, \mathcal{C})$ that approximates $\log p_{\boldsymbol{\theta}}(\cdot|\mathcal{C})$. This completes the proof. $\qquad\square$

### A.2    THEORETICAL ISSUES ON EQ. (5)

We continue from Section 3.3 to scrutinize the theoretical challenges that arise in the naively-extended optimization framework in Eq. (5). To proceed, we first restate the objective function in Eq. (5):

$$\mathcal{J}(z_t, \mathcal{C}_{\boldsymbol{v}}) := \mathbb{E}_{\boldsymbol{\epsilon}}\big[\|\hat{z}_0^w(z_t, \mathcal{C}_{\boldsymbol{v}}) - \texttt{sg}(\hat{z}_0^w(z_{s|t,0}^w, \mathcal{C}_{\boldsymbol{v}}))\|_2^2\big].$$

Remember that we identified three theoretical issues that impair the connection to the target log-likelihood $\log p_{\boldsymbol{\theta}}(z_0 \mid \mathcal{C})$: (i) the reliance on the CFG-based clean predictions; (ii) obstructed gradient flow through the second term in the squared-L2 loss; and (iii) the incorporation of $\mathcal{C}_{\boldsymbol{v}}$ within the second term in the loss.

**CFG-based clean prediction.** We start by examining the first point, the pathology due to the CFG-based clean predictions. Suppose we incorporate the CFG-based clean predictions $\hat{z}_0^w$ in our framework Eq. (6), in place of the non-CFG terms $\hat{z}_0$. The objective function then becomes:

$$\mathcal{J}_{\mathcal{C}}^w(z_t, \mathcal{C}_{\boldsymbol{v}}) := \mathbb{E}_{\boldsymbol{\epsilon}}\big[\|\hat{z}_0^w(z_t, \mathcal{C}_{\boldsymbol{v}}) - \hat{z}_0^w(z_{s|t,0}^w, \mathcal{C})\|_2^2\big].$$

To see its connection to log-likelihood, let us consider the weighted sum of this objective with $\bar{w}_s :=$ $\alpha_s/(1 - \alpha_s)$ (as in Proposition 1). Manipulating the averaged objective similarly as in Section A.1 then yields:

$$
\sum_{s=1}^{T} \bar{w}_s \mathcal{J}_{\mathcal{C}}^{w}(\boldsymbol{z}_t, \mathcal{C}_{\boldsymbol{v}}) = \sum_{s=1}^{T} \bar{w}_s \mathbb{E}_{\boldsymbol{\epsilon}} \left[ \| \hat{\boldsymbol{z}}_0^{w}(\boldsymbol{z}_t, \mathcal{C}_{\boldsymbol{v}}) - \hat{\boldsymbol{z}}_0^{w}(\boldsymbol{z}_{s|t,0}^{w}, \mathcal{C}) \|_2^2 \right]
$$

$$
= \sum_{s=1}^{T} \frac{\alpha_s}{1 - \alpha_s} \mathbb{E}_{\boldsymbol{\epsilon}} \left[ \left\| \frac{1}{\sqrt{\alpha_s}} (\boldsymbol{z}_{s|t,0}^{w} - \sqrt{1 - \alpha_s} \boldsymbol{\epsilon}) - \frac{1}{\sqrt{\alpha_s}} (\boldsymbol{z}_{s|t,0}^{w} - \sqrt{1 - \alpha_s} \tilde{\boldsymbol{\epsilon}}_{\boldsymbol{\theta}}^{w}(\boldsymbol{z}_{s|t,0}^{w}, \mathcal{C})) \right\|_2^2 \right]
$$

$$
= \sum_{s=1}^{T} \mathbb{E}_{\boldsymbol{\epsilon}} [\| \boldsymbol{\epsilon} - \tilde{\boldsymbol{\epsilon}}_{\boldsymbol{\theta}}^{w}(\sqrt{\alpha_s} \hat{\boldsymbol{z}}_0^{w}(\boldsymbol{z}_t, \mathcal{C}_{\boldsymbol{v}}) + \sqrt{1 - \alpha_s} \boldsymbol{\epsilon}, \mathcal{C}) \|_2^2]. \tag{11}
$$

Observe that in the RHS of Eq. (11), we see the CFG noise estimation term $\tilde{\boldsymbol{\epsilon}}_{\boldsymbol{\theta}}^{w}$, instead of $\boldsymbol{\epsilon}_{\boldsymbol{\theta}}$ as in Eq. (10). This comes from the use of $\hat{\boldsymbol{z}}_0^{w}(\boldsymbol{z}_{s|t,0}^{w}, \mathcal{C})$ in the second term of the squared-L2 loss. Since $\tilde{\boldsymbol{\epsilon}}_{\boldsymbol{\theta}}^{w}$ represents a distinct probability density, say $\tilde{p}_{\boldsymbol{\theta}}(\cdot \mid \mathcal{C})$, the averaged objective in Eq. (11) is no longer connected to our focused conditional log-likelihood $\log p_{\boldsymbol{\theta}}(\cdot \mid \mathcal{C})$.

One may wonder whether the use of CFG for the first term in the squared-L2 loss of Eq. (6) is safe. However, we claim that it is also problematic. To show this, we derive the associated log-likelihood, which is immediate with the algebra used for Eq. (11):

$$
\sum_{s=1}^{T} \bar{w}_s \mathbb{E}_{\boldsymbol{\epsilon}} \left[ \| \hat{\boldsymbol{z}}_0^{w}(\boldsymbol{z}_t, \mathcal{C}_{\boldsymbol{v}}) - \hat{\boldsymbol{z}}_0(\boldsymbol{z}_{s|t,0}^{w}, \mathcal{C}) \|_2^2 \right] \gtrapprox -\log p_{\boldsymbol{\theta}}(\hat{\boldsymbol{z}}_0^{w}(\boldsymbol{z}_t, \mathcal{C}_{\boldsymbol{v}}) \mid \mathcal{C}).
$$

We see that now the diffusion model (represented by $p_{\boldsymbol{\theta}}$) should estimate the conditional log-density w.r.t. the CFG clean prediction $\hat{\boldsymbol{z}}_0^{w}(\boldsymbol{z}_t, \mathcal{C}_{\boldsymbol{v}})$. We argue that this estimation may be inaccurate, since the CFG clean sample in the T2I context is potentially off-manifold. As analyzed in Chung et al. (2024), the CFG clean prediction $\hat{\boldsymbol{z}}_0^{w}(\boldsymbol{z}_t, \mathcal{C}_{\boldsymbol{v}})$ is in fact an extrapolation between $\hat{\boldsymbol{z}}_0(\boldsymbol{z}_t, \mathcal{C}_{\boldsymbol{v}})$ and $\hat{\boldsymbol{z}}_0(\boldsymbol{z}_t)$ (controlled by $w$). As a result, it may deviate from the data manifold, particularly for high $w$ values commonly used in standard T2I scenarios; see Figure 3 in Chung et al. (2024) for details. This off-manifold issue is especially pronounced during the initial phase of inference, as also reported in other studies (Kynkäänniemi et al., 2024). See Table 4 for experimental results that support this claim.

**Obstructed gradient.** Now we move onto the second issue. From the above analysis, we saw that the noise prediction in the second term is crucial for relating the objective function to the log-likelihood, meaning that allowing gradient flow through the second term is essential for accurate likelihood optimization. However, blocking the gradient via the stop-gradient on the second term contradicts this theoretical intuition. We found that the use of stop-gradient actually degrades performance; see Table 4 for instance.

$\mathcal{C}_{\boldsymbol{v}}$ **in the second term.** The reasoning behind the third challenge follows naturally from the previous analyses. In this case, the corresponding log-likelihood term can be derived as:

$$
\sum_{s=1}^{T} \bar{w}_s \mathbb{E}_{\boldsymbol{\epsilon}} \left[ \| \hat{\boldsymbol{z}}(\boldsymbol{z}_t, \mathcal{C}_{\boldsymbol{v}}) - \hat{\boldsymbol{z}}_0(\boldsymbol{z}_{s|t,0}, \mathcal{C}_{\boldsymbol{v}}) \|_2^2 \right] \gtrapprox -\log p_{\boldsymbol{\theta}}(\hat{\boldsymbol{z}}_0(\boldsymbol{z}_t, \mathcal{C}_{\boldsymbol{v}}) \mid \mathcal{C}_{\boldsymbol{v}}).
$$

We see that $\mathcal{C}_{\boldsymbol{v}}$ appears in conditioning variable, which diverges from our interest of approximating $\log p_{\boldsymbol{\theta}}(\cdot \mid \mathcal{C})$. See Table 4 for experimental results that corroborate this.

# B    SUPPLEMENTARY DETAILS

## B.1    DETAILS ON THE METRIC IN UM & YE (2024)

We continue from Section 3.3 to provide additional details on the likelihood metric developed by Um & Ye (2024). This original version is defined on pixel space $\boldsymbol{x}_0 \in \mathbb{R}^d$ (rather than latent domain $\boldsymbol{z}_0 \in \mathbb{R}^k$ as ours), formally written as (Um & Ye, 2024):

$$
\mathcal{J}(\boldsymbol{x}_t; s) := \mathbb{E}_{\boldsymbol{\epsilon}} \left[ d(\hat{\boldsymbol{x}}_0(\boldsymbol{x}_t), \hat{\boldsymbol{x}}_0(\boldsymbol{x}_{s|t,0})) \right],
$$

where $\boldsymbol{x}_t$ is a noisy pixel-domain image, and $\hat{\boldsymbol{x}}_0(\boldsymbol{x}_t)$ represents a clean estimate of $\boldsymbol{x}_t$: $\hat{\boldsymbol{x}}_0(\boldsymbol{x}_t) :=$ $(\boldsymbol{x}_t - \sqrt{1-\alpha_t}\boldsymbol{\epsilon}'_{\boldsymbol{\theta}}(\boldsymbol{x}_t))/\sqrt{\alpha_t}$, where $\boldsymbol{\epsilon}'_{\boldsymbol{\theta}}$ denotes a pixel diffusion model (different from our $\boldsymbol{\epsilon}_{\boldsymbol{\theta}}$). Here $\boldsymbol{x}_{s|t,0}$ indicates a noised version of $\hat{\boldsymbol{x}}_0(\boldsymbol{x}_t)$ according to timestep $s$: $\boldsymbol{x}_{s|t,0} := \sqrt{\alpha_s}\hat{\boldsymbol{x}}_0(\boldsymbol{x}_t) +$ $\sqrt{1-\alpha_s}\boldsymbol{\epsilon}$, and $\hat{\boldsymbol{x}}_0(\boldsymbol{x}_{s|t,0})$ is a denoised version of $\boldsymbol{x}_{s|t,0}$. $d$ indicates a discrepancy metric (*e.g.*, LPIPS (Zhang et al., 2018)). This quantity is interpretable as a reconstruction loss of $\hat{\boldsymbol{x}}_0(\boldsymbol{x}_t)$, and theoretically, it is an estimator of the negative log-likelihood of $\hat{\boldsymbol{x}}_0(\boldsymbol{x}_t)$ (Um & Ye, 2024).

Similar to ours, the authors in Um & Ye (2024) employs this metric as a guidance function for minority sampling, sharing similar spirit as ours. In doing so, they propose several techniques such as stop-gradient, learning-rate scheduling, and the incorporation of LPIPS as $d$. Their proposed metric for the guidance function is expressible as:

$$\mathcal{J}(\boldsymbol{x}_t; s) := \eta_t \mathbb{E}_{\boldsymbol{\epsilon}}\big[\text{LPIPS}(\hat{\boldsymbol{x}}_0(\boldsymbol{x}_t), \text{sg}(\hat{\boldsymbol{x}}_0(\boldsymbol{x}_{s|t,0})))\big],$$

where $\eta_t$ indicates learning rate at time $t$ designed to decrease over time, and LPIPS is the perceptual metric proposed by Zhang et al. (2018). Although this approach offers considerable advantages in traditional image generation tasks (such as unconditional generation), it is not optimized for T2I generation, which presents unique challenges and requires more specialized techniques. This is confirmed by our experimental results, where a straightforward extension of their framework yields only modest performance improvements. See Table 2b for details.

## B.2 IMPLEMENTATION DETAILS

**Pretrained models and baselines.** We employed the official checkpoints provided in HuggingFace for all three pretrained models. For the null-prompted DDIM baselines, we employed "commonly-looking" as the null-text prompt for all three pretrained models. The CADS baselines were primarily obtained using the recommended settings in the paper (Sadat et al., 2023), while we adjusted the hyperparameters on SDXL-Lightning for adaptation to distilled models. Specifically, we set $\tau_1 = 0.8$, $\tau_2 = 1.0$, and $s = 0.1$, while keeping other settings unchanged. For SGMS, we respected the original design choices (like the use of sg) and tuned the remaining hyperparameters to attain the optimal performance in the T2I context. In particular, we used the squared-L2 loss as the discrepancy metric and employed $s = 0.75T$. For their latent optimizations, we employed Adam optimizer (Kingma, 2014) (as ours) with learning rates between 0.005 and 0.01. Similar to ours, latent updates were performed intermittently, with $N = 3$ (*i.e.*, one update per three sampling steps). Each latent optimization consisted of three distinct update steps: $K = 3$.

**Evaluations.** The ClipScore values reported in our paper were due to torchmetrics[3]. For PickScore and Image-Reward, we employed the implementations provided in the official code repositories[4][5]. Precision and Recall were computed with $k = 5$ using the official codebase of Han et al. (2022)[6]. The log-likelihood values were evaluated based on the implementation of Hong et al. (2024)[7]. In-Batch Similarity that we used in the diversity optimization (in Table 3b) were computed with the repository of Corso et al. (2023)[8].

**Hyperparameters.** Our results were obtained using $s = T - t$, and we used Adam optimizer with $K = 3$, similar to SGMS. Learning rates were set between 0.001 and 0.002 across all experiments. We shared the same intermittent update rate of $N = 3$ with SGMS. For initializing $v$, we shared the same word embedding for "cool" for the main results (presented in Table 1). The number of learnable tokens for our approaches was set to 1. As described in Section 3.3, we globally used $\lambda = 1$ across all experiments. For the experiments on SDXL-Lightning that involves two distinct text-encoders, we employed a single Adam optimizer to jointly update both embedding spaces to minimize parameter complexity. We also synchronized other design choices for the two encoders, *e.g.*, sharing the same initial token embedding.

---

[3]https://lightning.ai/docs/torchmetrics/stable/multimodal/clip_score.html
[4]https://github.com/yuvalkirstain/PickScore
[5]https://github.com/THUDM/ImageReward
[6]https://github.com/hichoe95/Rarity-Score
[7]https://github.com/unified-metric/unified_metric
[8]https://github.com/gcorso/particle-guidance

| Method | CLIPScore ↑ | PickScore ↑ | ImageReward ↑ | Precision ↑ | Recall ↑ | Likelihood ↓ |
|---|---|---|---|---|---|---|
| DDIM | 31.4395 | **21.4570** | 0.1845 | **0.6070** | 0.7094 | 1.0465 |
| Eq. (6) (proposed) | **31.7369** | 21.3522 | **0.2839** | 0.5420 | **0.7340** | **0.9230** |
| Eq. (6) + $\hat{z}_0^w$ | 30.5193 | 20.7307 | -0.1468 | 0.4890 | 0.7182 | 0.9399 |
| Eq. (6) + $\mathtt{sg}$ | 31.6597 | 21.3114 | 0.2738 | 0.5230 | 0.7284 | 0.9290 |
| Eq. (6) + $\mathcal{C}_v$ | 31.6676 | 21.3652 | 0.2808 | 0.5550 | 0.7262 | 0.9281 |
| Eq. (6) + all (*i.e.*, Eq. (5)) | 30.2994 | 20.4840 | -0.1944 | 0.4760 | 0.6864 | 0.9245 |

Table 4: **Impacts of theoretical flaws in Eq. (5).** "+ $\hat{z}_0^w$" indicates the case that further incorporates the CFG clean predictions into Eq. (6). "+ $\mathtt{sg}$" refers to the one employing the stop-gradient on $\hat{z}_0(z_{s|t,0}, \mathcal{C})$. "+ $\mathcal{C}_v$" represents the setting of feeding $\mathcal{C}_v$ in the computations of $\hat{z}_0(z_{s|t,0}, \mathcal{C})$ in place of $\mathcal{C}$. "+ all" is the case that employs all the above three flawed choices, *i.e.*, Eq. (5). We observe clear performance benefits of our theory-driven design choices over the naive framework in Eq. (5). The results were obtained on SDv1.5.

| Target | CS ↑ | LL ↓ |
|---|---|---|
| Eq. (6) | 31.3658 | 0.5449 |
| Eq. (7) | **31.4194** | 0.5449 |

(a) Influence of $\mathtt{sg}$-trick

| Type | CS ↑ | LL ↓ |
|---|---|---|
| $s = 0.75T$ | 31.4534 | 0.9469 |
| $s = T - t$ | **31.7369** | **0.9230** |

(b) Impact of adaptive $s$

| Type | CS ↑ | LL ↓ |
|---|---|---|
| $\mathcal{C}$ | 31.7548 | 0.9744 |
| $\mathcal{C}_{v^*}$ | **31.7871** | **0.9511** |

(c) Effect of using $\mathcal{C}$

Table 5: **Effectiveness of our new techniques.** "CS" denotes ClipScore (Hessel et al., 2021), while 'LL' indicates log-likelihood. "$\mathcal{C}$" refers to the use of $\mathcal{C}$ during sampling steps without prompt optimization (when incorporating an intermittent prompt update, *i.e.*, $N > 1$). On the other hand, "$\mathcal{C}_{v^*}$" refers to the use of optimized token embeddings in the latest steps. Our results show that the proposed design choices consistently outperform naive approaches. The results in (a) were obtained using SDXL-Lightning, while SDv1.5 was employed for (b) and (c).

**Computational complexity.** The inference time for DDIM is approximately 1.136 seconds per sample, with CADS requiring a similar amount of time. The complexities of SGMS and our approach are rather higher due to the inclusion of backpropagation and iterative updates of latents or prompts. Specifically, SGMS takes 5.756 seconds per sample, while our sampler requires slightly more time – 6.205 seconds per sample – which we attribute to the additional backpropagation pass introduced by our removal of gradient-blocking. All computations herein were performed on SDv2.0 using a single NVIDIA A100 GPU.

**Other details.** Our implementation is based on PyTorch Paszke et al. (2019), and experiments were performed on twin NVIDIA A100 GPUs. Code is available at `https://github.com/anonymous-6898/MinorityPrompt`. For

## C  ADDITIONAL ABLATIONS AND DISCUSSIONS

### C.1  FURTHER ABLATION STUDIES

Table 4 exhibits the individual impacts of the three theoretical flaws in the naive framework in Eq. (5). We highlight that our new design choices motivated by a set of careful theoretical analyses yields significant advantages specifically in preserving text-alignment and user-preference. This further validates our framework as a powerful minority sampler that achieves high-quality generation.

Table 5 explores the impact of our techniques developed for further improvements in Section 3.3. We see consistent enhancements over naive design choices. A key insight from Table 5c is that reusing token embeddings optimized at earlier timesteps, denoted as "$\mathcal{C}_{v^*}$" in the table, offers limited benefit compared to simply using the base prompts $\mathcal{C}$. This finding highlights the evolutionary nature of our prompt-tuning framework, which supports continual updates to embeddings across sampling timesteps.

Table 6 investigates the design choices related to learnable tokens in our framework. Observe in Table 6a that our framework consistently delivers significant performance gains across different initial

| Init word | CS ↑ | LL ↓ |
|---|---|---|
| "uncommon" | 31.6971 | **0.8868** |
| "special" | 31.6178 | 0.9342 |
| "cool" | **31.7369** | 0.9230 |

(a) Sensitivity to the initial word

| Position | CS ↑ | LL ↓ |
|---|---|---|
| – | 31.4395 | 1.0465 |
| Prefix | 31.5519 | 0.9249 |
| Postfix | **31.7369** | **0.9230** |

(b) Impact of the position of $\mathcal{S}$

| # of tokens | CS ↑ | LL ↓ |
|---|---|---|
| 1 | **31.6465** | **0.9006** |
| 2 | 31.5866 | 0.9163 |
| 4 | 31.4989 | 0.9419 |

(c) Effect of # of learnable tokens

Table 6: **Exploring the design space of learnable tokens.** "Init word" indicates the word embedding used for initializing $v$. "–" refers to standard DDIM sampling without prompt optimization. "Prefix" denotes prepending the placeholder string $\mathcal{S}$ to $\mathcal{P}$, while "Postfix" indicates appending it to the end of $\mathcal{P}$. "# of tokens" represents the number of tokens assigned to the string $\mathcal{S}$. We observe that the proposed approach is not highly sensitive to the choice of initial word, and as suggested, attaching $\mathcal{S}$ at the end of the prompts yields the best performance. Additionally, using a single token is sufficient to achieve performance gains. We used SDv1.5 for the results herein.

| Method | CLIPScore ↑ | PickScore ↑ | ImageReward ↑ | Precision ↑ | Recall ↑ | Likelihood ↓ |
|---|---|---|---|---|---|---|
| DDIM | 31.4395 | **21.4570** | 0.1845 | **0.6070** | 0.7094 | 1.0465 |
| DDIM-CFG++ | 31.4755 | 21.4490 | 0.1938 | 0.5710 | 0.7100 | 1.0452 |
| Ours | 31.7369 | 21.3522 | 0.2839 | 0.5420 | **0.7340** | 0.9230 |
| Ours-CFG++ | **31.7627** | 21.3399 | **0.3062** | 0.5540 | 0.7284 | **0.9183** |

Table 7: **Compatibility with other sampling techniques.** "DDIM-CFG++" represents the DDIM sampler integrated with CFG++ (Chung et al., 2024), while "Ours-CFG++" is our MinorityPrompt framework implemented with CFG++. We highlight that MinorityPrompt demonstrates strong compatibility, delivering significant performance gains even when combined with CFG++. The guidance weights for the CFG++ cases were set to 0.6, *i.e.*, the recommended setting in the paper (Chung et al., 2024). We used SDv1.5 for the results herein.

word embeddings. Regarding the position of $\mathcal{S}$, appending it to the end of the prompts yields better results. We speculate that prepending may have a greater impact on the semantics of the text embeddings due to the front-weighted nature of the training process for the CLIP text encoders (Radford et al., 2021) employed in our T2I models. As exhibited in Table 6c, a single token is sufficient to realize the performance benefits of our approach. The performance degradation observed with increasing tokens is likely due to their heightened influence on semantics, similar to the effect of $\mathcal{S}$'s position.

Table 7 investigates the performance of MinorityPrompt when integrated with CFG++ (Chung et al., 2024) across different sampling techniques. We see that MinorityPrompt consistently outperforms standard DDIM-CFG++ by a notable margin, demonstrating the robustness and adaptability of our approach. This compatibility with CFG++ highlights the flexibility of MinorityPrompt, enabling substantial gains even when leveraging advanced conditioning strategies.

## C.2 LIMITATIONS AND DISCUSSION

A disadvantage is that our framework introduces additional computational costs (similar to Um & Ye (2024)), particularly when compared to standard samplers like DDIM. As noted in Section B.2, this is mainly due to the incorporation of backpropagation and iterative updates of prompts. Additionally, the removal of gradient-blocking, aimed at restoring the theoretical connection to the target conditional density, further contributes to the overhead. Future work could focus on optimizing these processes to reduce computational demands. One potential approach is to develop an approximation of our objective that mitigates the need for extensive backpropagation while maintaining its alignment with the target log-likelihood.

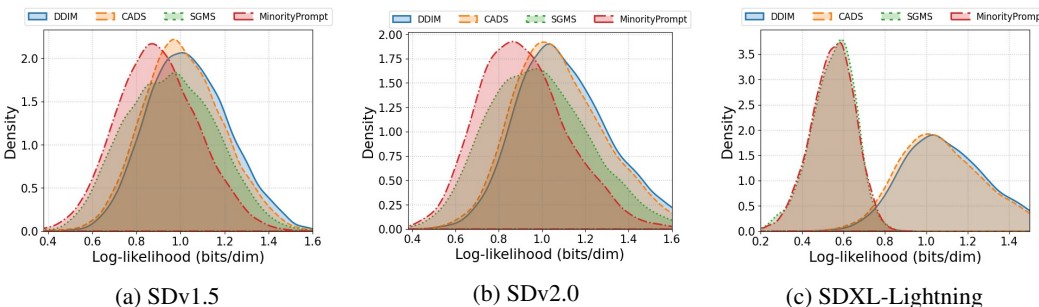

(a) SDv1.5        (b) SDv2.0        (c) SDXL-Lightning

Figure 5: **Comparison of log-likelihood distributions.** The likelihood values were measured using the PF-ODE-based computation proposed by Song et al. (2020b). We observe that MinorityPrompt better produces low-likelihood instances compared to the considered baselines across all three pretrained models.

# D  ADDITIONAL EXPERIMENTAL RESULTS

## D.1  LOG-LIKELIHOOD DISTRIBUTIONS

Figure 5 exhibits the log-likelihood distributions for MinorityPrompt and the baseline models across all three pretrained architectures. We see that MinorityPrompt consistently produces lower log-likelihood instances, further demonstrating its improved capability of generating minority samples. The distributions for SDXL-Lightning are more dispersed than in other scenarios, which may be attributed to the larger latent space upon which SDXL-Lightning is based. The competitive results compared to SGMS observed in SDXL-Lightning may arise from the limited optimization opportunities available in distilled models (as discussed in the manuscript).

## D.2  ADDITIONAL GENERATED SAMPLES

To facilitate a more comprehensive qualitative comparison among the samplers, we provide an extensive showcase of generated samples for all the focused T2I pretrained models. See Figures 6–8 for details.

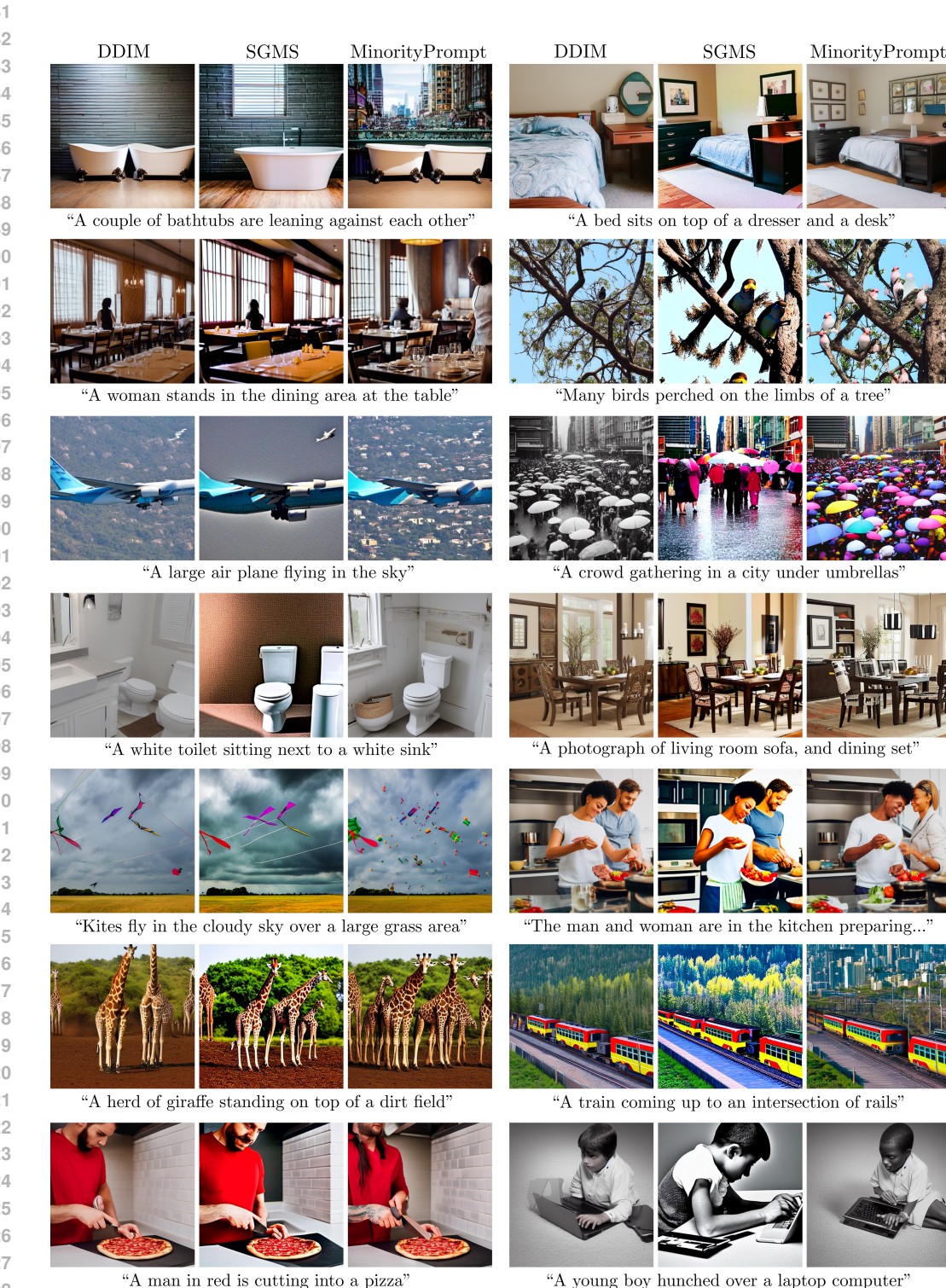

DDIM     SGMS     MinorityPrompt     DDIM     SGMS     MinorityPrompt

"A couple of bathtubs are leaning against each other"     "A bed sits on top of a dresser and a desk"

"A woman stands in the dining area at the table"     "Many birds perched on the limbs of a tree"

"A large air plane flying in the sky"     "A crowd gathering in a city under umbrellas"

"A white toilet sitting next to a white sink"     "A photograph of living room sofa, and dining set"

"Kites fly in the cloudy sky over a large grass area"     "The man and woman are in the kitchen preparing..."

"A herd of giraffe standing on top of a dirt field"     "A train coming up to an intersection of rails"

"A man in red is cutting into a pizza"     "A young boy hunched over a laptop computer"

Figure 6: **Generated samples on SDv1.5.** Generated samples from three distinct samplers: (i) DDIM (Song et al., 2020a); (ii) SGMS (Um & Ye, 2024); (iii) MinorityPrompt (ours). Random seeds were shared across all three methods.

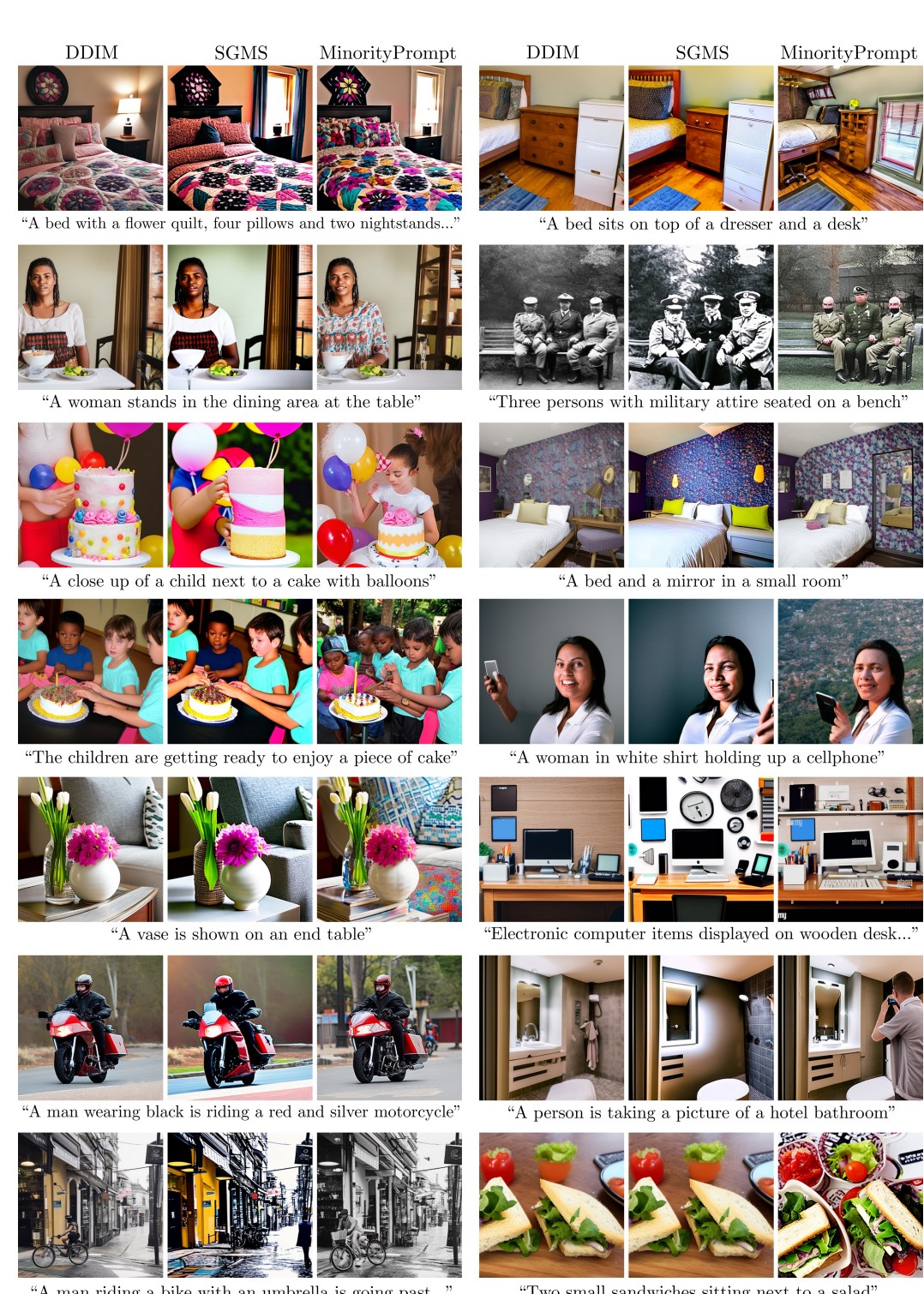

Figure 7: **Generated samples on SDv2.0.** Generated instances from three different techniques: (i) DDIM (Song et al., 2020a); (ii) SGMS (Um & Ye, 2024); (iii) MinorityPrompt (ours). We shared the same random seeds across all three approaches.

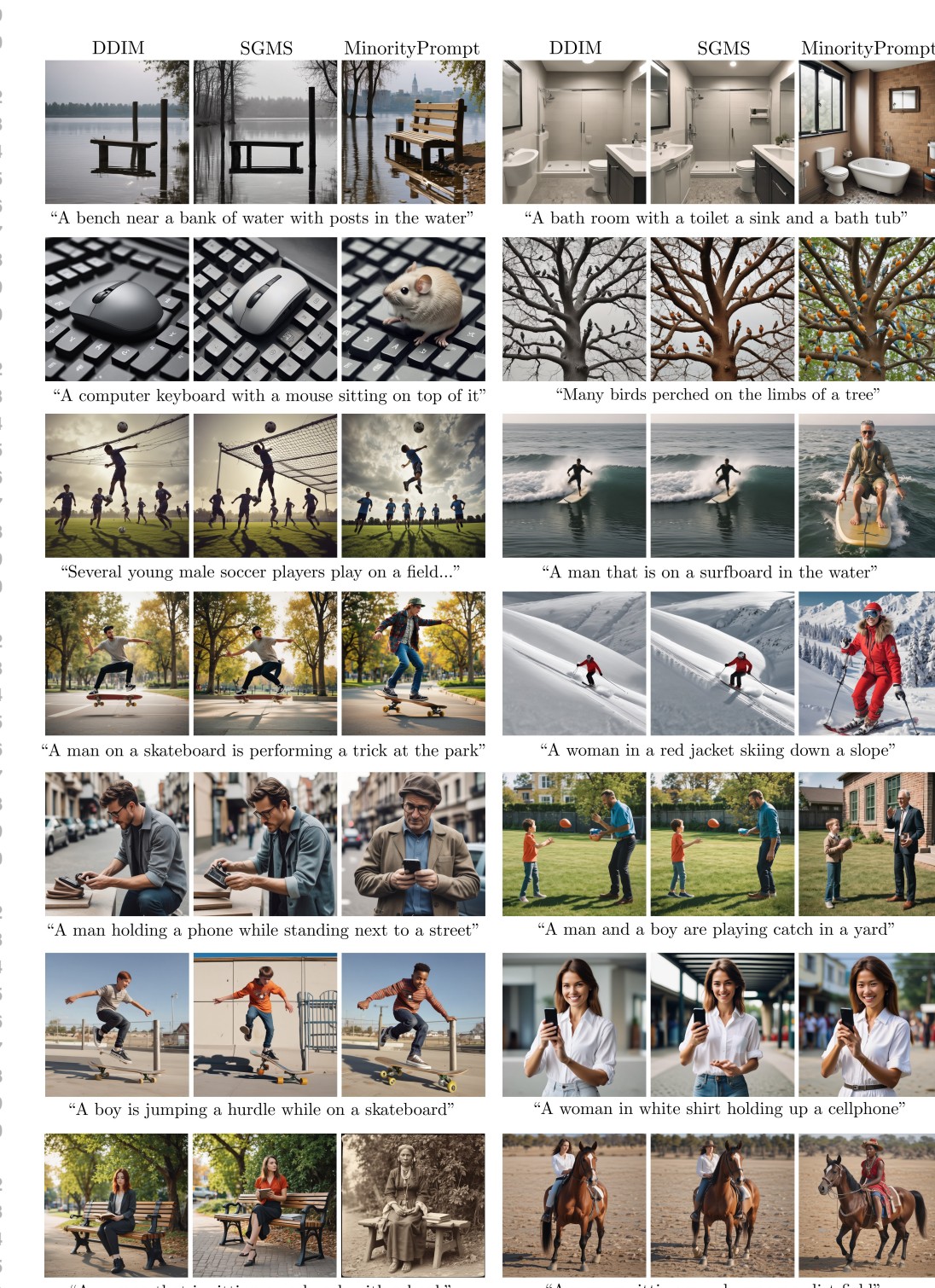

Figure 8: **Additional generated samples on SDXL-Lightning.** Generated samples from three different approaches: (i) DDIM (Song et al., 2020a); (ii) SGMS (Um & Ye, 2024); (iii) MinorityPrompt (ours). We employed the same initial noises across all three samplers.

