# OpenReview forum: "MinorityPrompt: Text to Minority Image Generation via Prompt Optimization"
_ICLR.cc/2025/Conference — ICLR 2025 Conference Withdrawn Submission_

### Official Review · Reviewer_8uFC · 2024-10-27

**Soundness:** 3
**Presentation:** 2
**Contribution:** 2
**Rating:** 3
**Confidence:** 3

**Summary:**

This paper investigates the generation of minority and uncommon samples using pre-trained text-to-image diffusion models. The authors propose a framework to shift the focus of these models from high-density regions towards areas of lower density by minimizing a likelihood metric tailored to capture the uniqueness of noisy intermediate samples. This is done by optimzing a new token embedding on the fly Additionally, they present techniques to enhance both the quality of generated results and semantic controllability. Qualitative and quantitative comparisons were conducted across three different diffusion models to demonstrate the effectiveness of their approach.

**Strengths:**

- The paper is well orginized and easy to follow.
- The idea of optimizing a single toke embedding to preserve the intended semantics while generating minority features is interesting.

**Weaknesses:**

- Limited Novelty: The authors adapt an existing idea from [1] for text-to-image models, adding techniques to enhance optimization stability and semantic controllability for generating minority images. However, the core concept remains similar to that of [1].
- No qualitative or quantitative comparison is provided against the proposed approach in Eq.(5). The authors argue that it has “theoretical issues that limit performance gains,” but there is no supporting evidence in the paper.
- DDIM+null seems like a strong baseline however no qualitative comparison is shown against it.
- The qualitative results in Table 1 are unconvincing. While the likelihood is lowest, the method mostly shows improved results against baselines in SD2.1. The lack of similar improvements in other diffusion models is not clear.
- How does the proposed method impact image quality? A systematic evaluation is needed to assess this.
- Precision and recall are known to be inadequate metrics for diversity evaluation. The authors should consider using [2] to assess their method.
- Why would CLIPScore improve for the proposed method if the text input remains unchanged?
- Following my last three comments, I suggest to conduct a user study to measure the diversity and quality of your approach compared to other baselines.
- Would optimizing more than one token lead to better results?
- On line 296, it’s noted that placing the placeholder string at the end of the prompt yields the best performance. Why might this be the case?

[1] Um., et al. (2024) *Self-guided generation of minority samples using diffusion models.*
[2] Naeem., et al. (2024) *Reliable Fidelity and Diversity Metrics for Generative Models.*

**Questions:**

See weaknesses. My biggest concern is the limited novelty as it is a relatively small incremental step of [1].

---

### Official Review · Reviewer_2FmW · 2024-10-31

**Soundness:** 3
**Presentation:** 3
**Contribution:** 2
**Rating:** 5
**Confidence:** 3

**Summary:**

This paper investigates the behavior of T2I model in low-density data distribution and proposes an online prompt optimization framework  to improve minority generation. Concretely, it injects a learnable token in the text encoder that is updated on the fly to maximize a carefully designed objective function to achieve the desired generation result.  Through extensive results, the author show that the proposed method can generate images with high quality and prompt alignment in low likelihood regions. The author also explored to use this method as a way to mitigate biase in pretrained T2I model

**Strengths:**

1. The presentation is clear and easy to follow. The paper provides clear intuition and motivations for the proposed objective by starting with a naive application of previous methods. It provides careful theoretical analysis on the weakness of this naive application (eq 5) and proposed a novel approach to address them.

2. The author provides extensive experiments on three models (SDv1.5,SDv2.0, SDXL-LT), demonstrating the proposed method can generalize to different model architectures.

**Weaknesses:**

1. The paper lacks analysis on the static significant of evaluation results. This is particularly relevant as the different metrics have varying scales. One of the author's major claim is that the proposed method achieves "reason generation quality" in "low likelihood" regime.  For example, the paper shows MinorityPrompt has 0.17 drop in PickScore on SDv1.5 (Table 1).  Its hard to contextualize if such difference should be considered as a major difference or minor difference without seeing the standard error or confidence interval. For example, if the stderr is +-0.2, then it means a statistical tie. If the stderr is less, then it may indicate that MinorityPrompt is worse than baseline with a sufficiently low p-value.

2. For the quantitative evaluation, it appears that MinorityPrompt often leads to higher prompt alignment (ClipScore) at the expense of image quality (PickScore). A similar tradeoff is oftentimes achieved through classifier free-guidance (CFG). The author uses a fixed CFG of 7.5. However, the author should vary the cfg of the base model and establish the frontier of ClipScore-ImageQuality tradeoff. It may be the case that MinorityPrompt is outside the frontier and is strictly better, or there may be a CFG that achieves higher ClipScore and PickScore than MinorityPriompt. Without this study, the results are inconclusive.

3. Experiments on Fairness are inconclusive, and the author fails to compare against baselines such as Iti-Gen[1],FairDiffusion[2], aDFT[3]. Use learnable token to achieve fair generation is not a novel idea. Hence, it is important to compare against existing literatures.

In the absence of human evaluation (which is understandable as they can be very costly), I would expect more discussion on these numerical metrics and how they translate into perceptual quality of generated images.

[1]Zhang, Cheng, et al. "Iti-gen: Inclusive text-to-image generation." Proceedings of the IEEE/CVF International Conference on Computer Vision. 2023.
[2]Friedrich, Felix, et al. "Fair diffusion: Instructing text-to-image generation models on fairness." arXiv preprint arXiv:2302.10893 (2023).
[3]Shen, Xudong, et al. "Finetuning text-to-image diffusion models for fairness." arXiv preprint arXiv:2311.07604 (2023).

**Questions:**

See weakness.
Overall, I find the paper well-motivated with good theoretical foundation. However, I find the current experiments failed to show the practical significance of the proposed method, especially since the statistic significance is not discussed and the paper uses a non-conventional benchmark. I would welcome responses that address weakness 1,2,3.

A few additional questions not mentioned in the weakness and not taken into the consideration of the decision:
1. The paper uses 10k images for SD1.5,2 and 5k for SDXL-LT. Why is this setup adopted? This is also relevant to weakness 1 as different number of samples will lead to different standard errors/confidence interval.
2. How is figure 4 generated?  Are samples randomly picked from different models? Or is the latent fixed.

---

### Official Review · Reviewer_i8hb · 2024-11-01

**Soundness:** 2
**Presentation:** 2
**Contribution:** 2
**Rating:** 5
**Confidence:** 3

**Summary:**

This paper proposes a method to generate more minority instances. The framework appends a trainable token after the prompt and optimizes this token in real-time during the sampling process. This approach aims to generate more minority instances while preserving the semantic integrity.

**Strengths:**

1. This paper proposes a prompt optimization method by placing a learnable token at the end of the sentence to preserve the original semantic information.
2. The paper explores a self-learning approach to optimize the prompt token, thereby enhancing the model's ability to generate more minority instances.
3. By setting different objective functions, more functionalities can be achieved.
4. The article is well-written, with clear and precise explanations.

**Weaknesses:**

1. For example, as mentioned in Fig. 1, there is ambiguity with biases such as 'man' with 'young'. Why can't we directly use prompt engineering methods like 'old man' as a prompt to solve the problem you mentioned?

2. It is necessary to use prompts corresponding to minority instances to generate images and observe the advantages of your method compared to existing methods. Without a detailed prompt, generating any image is reasonable, and I cannot consider it a minority instance scenario; it only indicates that the model tends to generate certain samples.

3. Additional experiments on different samplers (ODEs, SDEs) are needed to verify the effectiveness.

4. This paper introduces additional training, so how is the efficiency?

5. How about the performance changes in diffusion models with fewer steps?

**Questions:**

As shown in Weaknesses.

---

### Official Review · Reviewer_qRGy · 2024-11-11

**Soundness:** 3
**Presentation:** 3
**Contribution:** 3
**Rating:** 6
**Confidence:** 4

**Summary:**

This paper presents a method to enable text-to-image diffusion models to generate minority samples, those less common in training data. Specifically, an online prompt optimization framework is developed to encourage the emergence of desired properties by optimizing text embedding of learnable tokens. Subsequently, this framework is tailored into a specialized solver that promotes the generation of minority features by incorporating a carefully crafted likelihood objective.  Comprehensive experiments, conducted across various types of T2I models, demonstrate that the proposed approach significantly enhances the capability to produce high-quality minority instances compared to existing samplers.

**Strengths:**

1. The proposed method is validated with multiple text-to-image diffusion models, showing the generalizability across different models including distilled backbones such as SDXL-Lightning
2. The proposed method can effectively encourage the emergence of low-likelihood samples and can be applied to mitigate the bias issue of text-to-image diffusion models, as supported by the quantitative evaluation results.
3. The proposed method only needs to optimize for  learnable tokens, without affecting the semantics of the input text prompt and therefore can improve diversity without compromising text alignment and image quality too much
4. The manuscript presents detailed analysis and effective solutions for the issues of related work (Um & Ye, 2024)

**Weaknesses:**

The authors claim that the method improves the ability of creating minority samples with minimal compromise to image quality but there are no experimental results to support this point. It would make the manuscript stronger if the authors could add image quality analysis such as the FID comparisons.

**Questions:**

would it be possible to provide a quantitative analysis of how favoring low-density samples would affect image quality?

---

### Note · Authors · 2024-11-14

I have read and agree with the venue's withdrawal policy on behalf of myself and my co-authors.